# Identifying Critical Factors and Trends Leading to Fatal Accidents in Small-Scale Construction Sites in Korea

Jong-Moon Hwang [1], Jeong-Hun Won [2,3,4] , Hyeon-Ji Jeong [3] and Seung-Hyeon Shin [4,*]

[1] Occupational Safety and Health Research Institute, Korea Occupational Safety and Health Agency, Ulsan 44429, Republic of Korea; bm0722@kosha.or.kr

[2] Department of Safety Engineering, Chungbuk National University, Cheongju 28644, Republic of Korea; jhwon@chungbuk.ac.kr

[3] Department of Disaster Prevention Engineering, Chungbuk National University, Cheongju 28644, Republic of Korea; gus1895@naver.com

[4] Department of Big Data, Chungbuk National University, Cheongju 28644, Republic of Korea

\* Correspondence: shshin0317@chungbuk.ac.kr; Tel.: +82-10-5813-5959

**Abstract:** Small-scale construction sites in South Korea account for about 91.5% of all construction workplaces and contribute to 72.3% of the total accidents and fatalities. Safety measures at these sites are often underestimated, and proper safety education is lacking. In particular, the fatality rate is about 4.43 times higher compared to medium-/large-scale construction sites. In this study, a systematic analysis was conducted to examine the causes and trends of industrial accidents in small-scale construction sites to address these issues. This study analyzed industrial accidents in small-scale construction sites using statistical analysis, LDA topic modeling, and network analysis based on data from the Korea Occupational Safety and Health Agency (KOSHA) from 2018 to 2022. The analysis revealed that the most critical cause of accidents in small-scale construction sites is 'Scaffolding and working platforms', with accidents primarily involving 'Fall'. Furthermore, various risk factors and accident trends were identified in apartment construction, new building projects, and mobile scaffolding usage. This study systematically analyzed the causes and trends of industrial accidents at small-scale construction sites, providing important evidence to enhance safety management and preventive measures. The results are expected to play a crucial role in establishing a safety culture at construction sites and ensuring the wellbeing of construction workers.

**Keywords:** small-scale construction sites; safety management; accident causes; accident types; accident trends

## 1. Introduction

Infrastructure facilities such as roads, bridges, and buildings are essential for improving the quality of life for citizens and driving national economic development. Consequently, the construction industry is a core driving force behind a country's progress. The fact that the construction industry is both a crucial sector and entails high risks cannot be ignored [1]. Globally, construction workers constitute only about 7% of the total workforce across industries, yet they account for a significantly larger proportion of fatalities [2,3]. While global efforts by governments and researchers have led to decreased construction-related industrial accidents, aiming to improve the industry's hazardous image, the construction industry still experiences more fatal accidents than other industries [4,5]. In Korea, as depicted in Figure 1, from the years 2013 to 2022, the accident fatality rate in the construction industry was consistently higher compared to the manufacturing sector. On average, the fatality rate in the construction sector was 3.09 times higher than in the manufacturing sector. This disparity was even more pronounced in 2020, where the difference in fatality rates between the two sectors reached a staggering fourfold. Such a stark contrast underscores the urgent need for addressing safety concerns in the construction

sector and offers a baseline for understanding the safety advancements and best practices in the manufacturing sector that could potentially be adopted in construction. When comparing accident fatality rates across industries, the construction industry consistently exhibits higher fatality rates than the manufacturing sector. In 2020, the fatality rate in construction (2.00) was about four times higher than that of manufacturing (0.50). Analyzing the trend in fatality rates, while the manufacturing and overall industries have experienced continuous decreases, the fatality rate of the construction industry has been on the rise again since 2016. In 2013, for the first time since then, the fatality rate in construction surpassed 2.00.

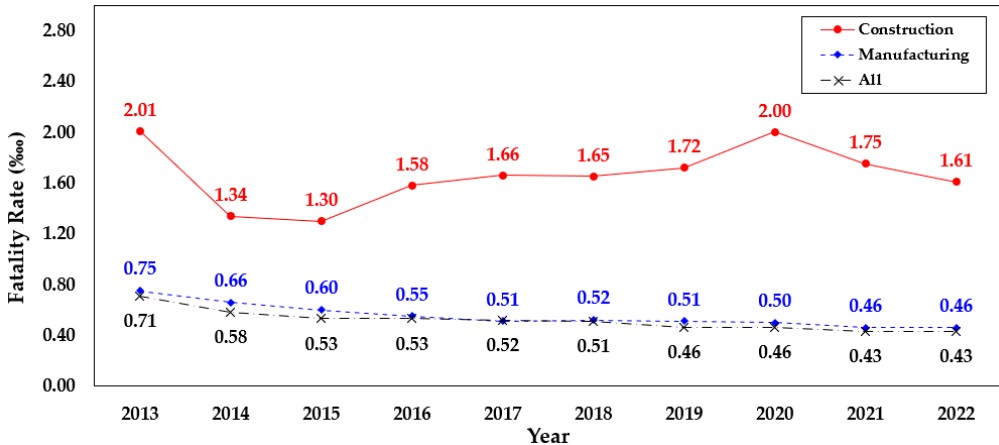

**Figure 1.** Work-related accident fatality rate across construction, manufacturing, and all industries in Korea.

The work-related accident fatality rate in the construction industry exhibits significant variations based on factors such as the number of on-site workers and the project costs. Workers at small-scale construction sites are exposed to higher accident risks than those at large-scale construction sites, a trend observed not only in developing countries but also in advanced countries [6–8]. In the case of South Korea, in 2020, there were 301,271 construction workplaces with construction costs of less than KRW 5 billion, constituting about 91.5% of all construction workplaces (329,279), and they accounted for about 72.3%, with 331 out of 458 fatalities. These data indicate that construction industrial accidents are concentrated in these smaller-scale areas (USD 1 ≈ KRW 1340). In particular, the fatality rate at small-scale construction sites is about 4.43 times higher than that of construction sites with project costs not less than KRW 12 billion. Thus, to effectively reduce industrial accidents in the construction industry, it is necessary to concentrate efforts on managing them in small-scale construction sites.

The various factors influencing the severity of fatal accidents occurring at small-scale construction sites can vary based on the safety and health regulations of the country, the culture, and the construction stakeholders. For instance, in the USA, the Occupational Safety and Health Administration (OSHA) sets and enforces standards to assure safe working conditions, particularly emphasizing training, outreach, and education. The UK's Health and Safety Executive (HSE) is focused on reducing work-related death and serious injury across all sectors, with a specific framework for construction that mandates specific safety roles and responsibilities for projects. In Australia, the Safe Work Australia body develops national policy relating to WHS (work health and safety) and workers' compensation, emphasizing consultation, cooperation, and coordination among various stakeholders in construction. However, according to multiple studies, small-scale construction projects are found to be facing common safety and health issues [5–8]; the significance of workplace safety measures is underestimated, and there is a lack of sufficient safety education for both workers and managers. In particular, construction clients often perceive industrial accidents as unpredictable events, leading to a tendency not to employ managers or experts with specialized knowledge in safety and health.

One of the key strategies to reduce industrial accidents at construction sites and enhance workplace safety is to conduct in-depth analyses of past incidents, identifying high-risk factors and implementing effective preventive measures to eliminate these risks [7]. Several countries, including South Korea, the United Kingdom, the United States, Japan, and Singapore, apply this methodology to decrease industrial accidents at small-scale construction sites. They periodically disclose industrial accident statistics and formulate safety and health management policies based on these data. Furthermore, various researchers analyze industrial accident cases to identify significant safety and health issues that the government may not recognize, presenting specific solutions to address them [7–9].

Most studies on improving safety and health at small-scale construction sites have recently focused on identifying the causes of accidents, their occurrences, and the populations of high-risk groups [7–11]. While these studies have significantly contributed to easily understanding the primary causes of accidents, they have overlooked the complex interactions between the complex causes and outcomes of accidents. In large-scale construction sites, the support of dedicated safety and health managers or experts enables the interpretation of and response to how complex accident causes interact and lead to accidents in a site-specific manner. However, in small-scale construction sites lacking specialized safety and health management support, simply providing accident causes is insufficient for understanding how the causes and outcomes interact to lead to accidents. Therefore, relying solely on statistical analysis may not be enough to prevent or respond to these incidents adequately. Therefore, in small-scale construction sites, providing specific accident scenarios explaining how the causes interact to trigger accidents can be more effective than simply listing the causes.

Furthermore, the existing research focused on small-scale construction sites in South Korea mostly relies on historical data, failing to capture the recent changes in work environments and trends. Therefore, the primary objective of this study is to systematically analyze the causes and trends of industrial accidents occurring in recent small-scale construction sites within South Korea. This aims to provide essential scientific evidence for establishing effective safety and health management strategies. In particular, this study identifies the critical causes and accident types that have the most significant impact on small-scale construction sites and presents how these causes and accident types actually lead to specific accidents.

The structure of this paper is as follows. Section 2, titled 'Background', delves into a comprehensive literature review of the existing research related to accidents in small-scale construction sites and explains the accident status in the domestic construction industry. In Section 3, the 'Materials and Methods' section is covered, including the data collection process for this study and the plan and methodology of this study. Next, in Section 4, the 'Results' are presented. A comparative analysis focusing on the causes and accident types was conducted between industrial accidents at small-scale construction sites with a construction cost of less than KRW 5 billion and medium-/large-scale construction sites. Additionally, the critical causes and accident types at small-scale construction sites were analyzed. For accidents related to the most critical cause and accident types at small-scale construction sites, LDA topic modeling and network analysis were conducted in this study, and the main results regarding accident trends were identified. Moving forward to Section 5, the 'Discussion' was carried out regarding the analysis results. Finally, in Section 6, the 'Conclusion' discusses the study's outcomes and outlines future research directions.

## 2. Background

### 2.1. Literature Review

#### 2.1.1. Accident Characteristics in Small-Scale Construction Sites

The recent literature primarily emphasizes the identification of accident causes, their frequency, and the groups most susceptible to these risks in small-scale construction sites. Cheng et al. [6] analyzed the characteristics of industrial accidents occurring at

small-scale construction sites in Taiwan. Dumrak et al. [7] found that 57.5% of all fatal accidents in south Australia happened at small-scale construction sites with less than 20 workers. Similarly, Camino López et al. [8] revealed that sites with fewer than 25 workers accounted for a staggering 58.1% of the total fatalities. Bang et al. [9] analyzed fatality rates based on construction cost and facility type using accidents from 2013 to 2019. The study revealed that serious accidents frequently occurred at sites of less than 0.008 billion USD in South Korea. Collectively, these studies, including contributions by Daba et al. [10] and Berhe et al. [11], consistently show that workers in smaller construction sites are at a higher risk compared to those in larger settings.

### 2.1.2. Contemporary Trends and Data Limitations in Construction Safety Research

While significant studies have been undertaken concerning accident characteristics in small-scale construction sites, there remains a gap in the literature concerning the specific differences between accidents in small-scale settings versus larger ones. Studies such as those by Lim et al. [5], Choi et al. [12], and Kang and Ryu [13] have predominantly focused on historical data, potentially overlooking contemporary shifts in work environments and trends. The reliance on pre-2019 data, given the escalating severity of recent industrial accidents in smaller construction sites, underscores the need for updated study methodologies and data sources.

### 2.1.3. The Role and Impact of Safety Training

Emerging studies have highlighted the pivotal role of safety training in enhancing hazard recognition and risk perception among construction workers. Fu et al. [14] underscored the potential of integrating visual cues in safety education to enhance hazard recognition abilities, especially among novice workers. Such innovative training approaches were further supported by Namian et al. [15], who emphasized the significance of high-engagement training methods in improving both hazard recognition and safety risk perception. Their findings revealed that high-engagement training led to superior hazard recognition and an elevated perception of safety risk. Furthermore, Perlman et al. [16] highlighted the discrepancies between how construction superintendents assess risk levels and the ratings provided by formal safety risk assessment methods. Their study accentuated the importance of training and education in bridging this gap. Moreover, Uddin et al. [17] identified specific hazard categories that construction workers are more proficient at recognizing, emphasizing the need for targeted training interventions.

In conclusion, while there has been considerable study on the characteristics and causes of accidents in small-scale construction sites, a profound need exists for innovative training methodologies that are rooted in a deep understanding of workers' hazard recognition patterns and challenges. The integration of technology, personalized learning interventions, and high-engagement training methods promises a more holistic and effective approach to construction safety education.

### 2.2. Accidents in Small-Scale Construction Sites in Korea

The safety and health management systems of construction sites in South Korea are based on the "Occupational Safety and Health Act", and different regulations apply depending on the scale of the construction project. The scale of construction is determined by the total construction cost, and generally, sites with construction costs less than KRW 5 billion are defined as small-scale construction sites. Currently, small-scale construction sites are mostly in blind spots regarding safety and health management regulations. Construction sites with construction costs of not less than KRW 5 billion and less than KRW 12 billion are considered medium-scale construction sites. They are subject to most safety and health management regulations, and safety managers must be appointed for on-site safety management. However, safety managers for medium-scale construction sites are allowed to have dual roles. On the other hand, construction sites with construction costs not less than KRW 12 billion are considered large-scale construction sites. Similar to

medium-scale sites, safety managers must be appointed, but safety managers for large-scale construction sites are dedicated safety managers and are not allowed to take on other duties outside of safety management works.

In the past 10 years, an analysis of the accident fatality rate due to accidents in the construction industry based on the construction cost revealed significant differences in South Korean construction sites (Figure 2). For construction sites with construction costs of less than KRW 12 billion, where there is no requirement for a dedicated safety manager, the fatality rate due to accidents was analyzed to be at least 2.5 times higher (in 2014) and up to 4.43 times higher (in 2020) compared to construction sites with construction costs of KRW 12 billion or more. The fatality rate in large-scale construction sites gradually increased from 2015 to 2019, which was followed by a decrease after 2020. Medium-scale construction sites exhibited a significant increase in 2016 followed by a continuous decrease until 2018, but the trend shifted to an increase in 2019. On the other hand, the fatality rate in small-scale construction sites showed repeated fluctuations until 2019, which was followed by a sharp increase in 2020 and a slight decrease in 2021.

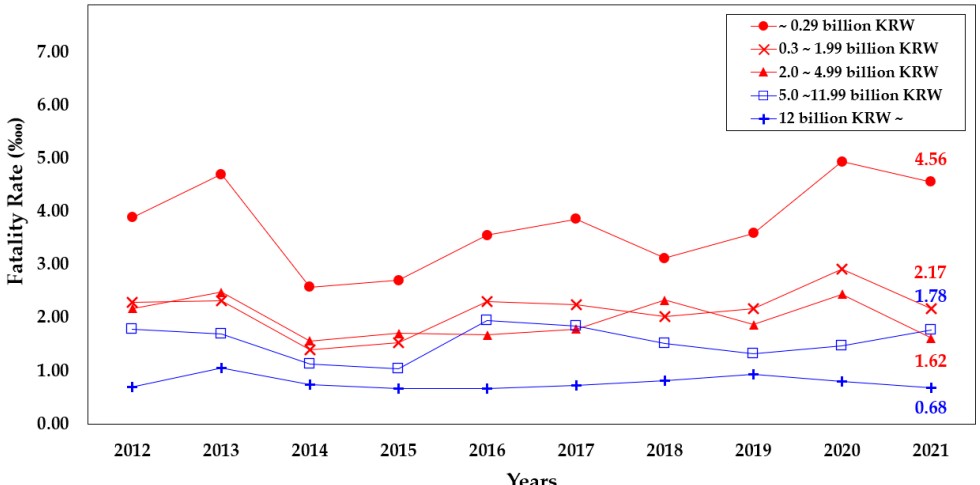

**Figure 2.** Work-related accident fatality rate according to construction costs in Korea.

## 3. Materials and Methods

### 3.1. Data Collection

In this study, accidents at small-scale construction sites were analyzed using the industrial accident status statistics data (referred to as "industrial accident data") reported to the Korea Occupational Safety and Health Agency (KOSHA) from 2018 to 2022. The industrial accident data include various fields such as industry category, construction scale, victim's name, diagnosis, cause, accident type, disability grade, country, occupation, date of death, age, gender, duration of employment, date of the accident, time of the accident, accident overview, employment status, and worker's position. Each field was prepared according to KOSHA's classification criteria.

In response to the importance of diverse data sourcing, it is worth mentioning that while our primary dataset originated from official sources, the inclusion of data from broader platforms, such as social media enterprises, could potentially enhance the comprehensiveness of the analysis. Such datasets, although rich in capturing real-time sentiments, can pose challenges related to data validity, consistency, and potential biases [18].

In this study, the analysis focused primarily on the cause and the accident type directly associated with industrial accidents. The classification criteria and definitions for the cause and the accident type are presented in Appendix A. In this study, any items or details within the causes and accident types that fell under other fields or with insufficient information were excluded from the analysis. Ultimately, 1511 cases of work-related fatalities due to accidents were analyzed in this study.

*3.2. Methods*

This study aimed to identify critical causes and accident types from industrial accident data in small-scale construction sites and to present trends in accidents associated with the most critical causes and accident types. This study involved several key steps, including data collection, statistical analysis, topic modeling, and network analysis. The research flowchart detailing these steps is presented in Figure 3.

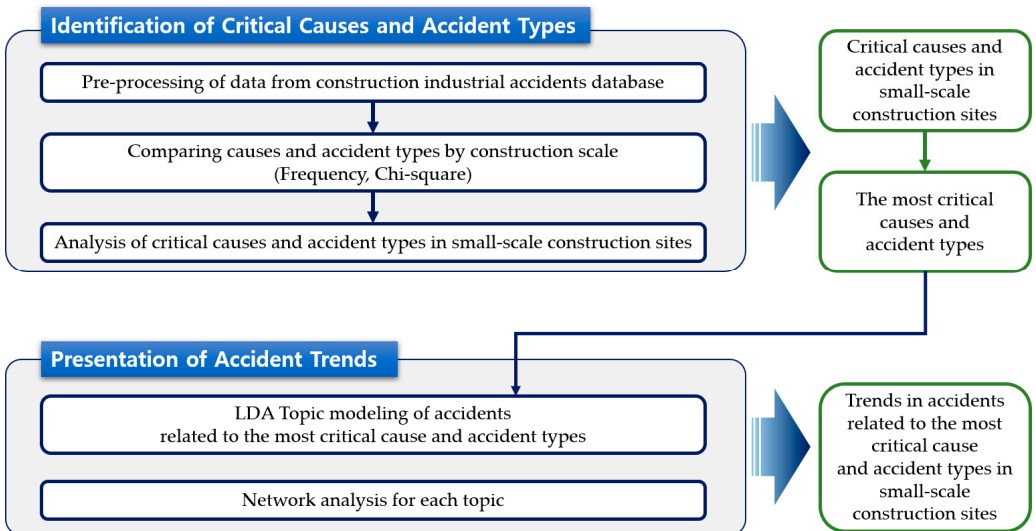

**Figure 3.** Research flowchart.

A chi-squared test was conducted to investigate the relationship of industrial accidents between small-scale construction sites and other construction sites in terms of causes and accident types. In this case, the chi-squared test was performed using the SPSS v26 software. If the chi-squared test indicated a difference in industrial accidents based on causes and accident types between small-scale construction sites and other construction sites, the critical causes and accident types for small-scale construction sites were identified using absolute and relative frequencies. In this case, the absolute frequency represents the frequency of accidents occurring in small-scale construction sites based on the causes or accident types. The relative frequency indicates the proportion of accident occurrences in small-scale construction sites compared to the total accident occurrences across all construction sites. The relative frequency is calculated according to Formula (1):

$$RF = (\text{Frequency at small-scale construction}/\text{Frequency at all construction}) \qquad (1)$$

RF represents the relative frequency, where Frequency at small-scale construction refers to the frequency of causes or accident types in small-scale construction sites, and Frequency at all construction refers to the frequency of causes or accident types across entire construction sites. Critical causes and accident types were presented in graph form by comprehensively considering absolute and relative frequencies. Through the analysis of the critical causes and accident types, it could be determined whether certain causes or accident types caused more accidents in small-scale construction sites relative to the total construction sites or if they were frequent in the entire construction industry. In the graph, causes or accident types located in the upper right corner frequently occurred in small-scale construction sites and were particularly hazardous, indicating that they pose a significant risk compared to the entire construction industry.

Next, to analyze specific trends in industrial accidents related to the critical causes and accident types identified in small-scale construction sites, latent Dirichlet allocation (LDA) topic modeling and network analysis were conducted. Topic modeling is a methodology that involves utilizing statistical and optimization algorithms to extract latent topics from

large text collections. It identifies topics that constitute the themes within an entire text collection, categorizing the text into relevant topics [19,20]. In topic modeling, the latent Dirichlet allocation (LDA) model is commonly used. The primary goal of LDA is to effectively identify latent topic information by analyzing the co-occurrence patterns of words within documents [21]. Despite the success and popularity of LDA, it has limitations in terms of capturing relationships between keywords derived from LDA in the way the text is represented [22]. Therefore, the study of Gerlach et al. [22] suggested that integrating topic modeling with network analysis could improve this aspect. Network analysis is a methodology that models complex systems as graphs composed of nodes and edges to analyze their structure and behavior. It is commonly used in data science to analyze complex relationships and patterns between nodes [23]. Therefore, this study analyzed the overview of industrial accidents caused by the most critical causes and accident types using LDA topic modeling to classify them into several accident types. In this case, the LDA topic modeling and network analysis were conducted using Python 3.9. Before conducting the topic modeling, the industrial accident overview data in the Korean natural language were tokenized as a preprocessing step. Initially, special characters, numbers, and characters other than Korean were removed. Tokenization was carried out using the Kkma morphological analyzer from the Konlpy package. Among the tokens obtained through tokenization, common Korean stopwords such as postpositional particles and conjunctions, along with proper nouns, terms related to hospital transfers, and other terms like confirmation, fact, and incident, which do not significantly impact accident analysis, were mostly removed as stopwords. After this preprocessing, the documents were transformed into a bag-of-words format to create a corpus. Next, the LDA topic modeling was performed using various packages from the Python gensim package, including LdaModel, Dictionary, MmCorpus, copora, models, TfidfModel, etc. The analysis to identify an appropriate number of topics was based on a comprehensive consideration of perplexity and coherence scores. Perplexity and coherence scores are quantifiable metrics that measure the predictability of a given set of texts. Generally, lower perplexity and higher coherence scores indicate a better prediction quality [24,25]. Subsequently, the LDA topic modeling was conducted based on the optimal number of topics. The network analysis using the network package was performed for each topic to analyze the complex interactions of accidents and extract accident trends. The network visualization was created using word pairs with the top 50 frequencies. The size of nodes and the thickness of edges were increased proportionally to the frequency of nodes and the connectivity of edges. Additionally, it's important to note that due to the proximity of certain keywords in the network visualization, there might be an overlap, causing certain keywords to be obscured. This visual overlap is a consequence of closely related keywords having a shorter distance between them in the network representation.

## 4. Results

### 4.1. Identification of Critical Causes and Accident Types

In this section, the critical causes and accident types leading to work-related fatal accidents in small-scale construction sites are presented. A comparative analysis was conducted on the causes and accident types of fatal accidents occurring in small-scale construction sites and other construction sites. Subsequently, the absolute frequency (AF) and relative frequency (RF) of accidents that occurred in small-scale construction sites were calculated to identify the critical causes and accident types contributing to fatal accidents.

#### 4.1.1. Comparing Causes and Accident Types by Construction Scale

The analysis examined the differences in industrial accidents based on causes and types of fatal accidents between small-scale construction sites and other construction sites. The results are summarized in Table 1. Among the 1511 work-related fatal accidents in the construction industry that were analyzed, there were 1013 accidents reported on small-scale construction sites with construction costs of less than KRW 5 billion, while 498 accidents occurred on medium-/large-scale construction sites with construction costs of not less

than KRW 5 billion. Work-related fatalities in small-scale construction sites accounted for about 67%.

**Table 1.** Chi-squared test and frequency analysis of accident causes and types in small-scale and other construction sites.

| Category | <KRW 50 Billion (*n*) | ≥KRW 50 Billion (*n*) | Total (*n*) |
|---|---|---|---|
| **Accident Causes** ($\chi^2$= 76.482, *p* < 0.001) | | | |
| Scaffolding and working platforms | 190 | 72 | 262 |
| Stepped structure and opening | 128 | 61 | 189 |
| Means of land transportation | 138 | 50 | 188 |
| Transport and lifting equipment/machinery | 99 | 81 | 180 |
| Construction/mining machinery | 81 | 66 | 147 |
| Stairs and ladders | 96 | 22 | 118 |
| General manufacturing and processing equipment/machinery | 73 | 28 | 101 |
| Floors, surfaces, etc. | 39 | 22 | 61 |
| Molds and supporting post | 24 | 31 | 55 |
| Electrical equipment, parts | 31 | 17 | 48 |
| Equipment/machinery, parts, and accessories | 19 | 15 | 34 |
| Materials | 23 | 7 | 30 |
| Components and accessories of buildings/structures | 15 | 10 | 25 |
| Humans, animals/plants | 15 | 2 | 17 |
| Portable power tools | 12 | 2 | 14 |
| Chemical products | 12 | 1 | 13 |
| Non-metallic mineral products | 8 | 1 | 9 |
| Containers, packaging and devices | 5 | 3 | 8 |
| Work environment, natural phenomena such as atmospheric conditions, etc. | 2 | 3 | 5 |
| Manual mechanical equipment | 2 | 2 | 4 |
| Hand tools | 1 | 0 | 1 |
| Fragments, debris, waste | 0 | 1 | 1 |
| Means of air, water transportation | 0 | 1 | 1 |
| Total | 1013 | 498 | 1511 |
| **Accident Types** ($\chi^2$= 39.582, *p* < 0.001) | | | |
| Fall | 547 | 223 | 770 |
| Collision | 109 | 67 | 176 |
| Struck by object | 59 | 60 | 119 |
| Pressed under/overturned | 73 | 30 | 103 |
| Caught in between | 45 | 30 | 75 |
| Collapse | 42 | 32 | 74 |
| Fire | 33 | 18 | 51 |
| Electrocution | 35 | 14 | 49 |
| Explosion/rupture | 24 | 5 | 29 |
| Chemical leakage | 14 | 4 | 18 |
| Tripping | 12 | 5 | 17 |
| Fallen in/Drown | 12 | 5 | 17 |
| Cutoff/cut/stab | 4 | 2 | 6 |
| Animal injury | 3 | 1 | 4 |
| Workplace traffic accident | 1 | 0 | 1 |
| Oxygen deficiency | 0 | 1 | 1 |
| Abnormal temperature contact | 0 | 1 | 1 |
| Total | 1013 | 498 | 1511 |

A chi-squared test was conducted to determine the statistical significance of the proportional differences in the causes of industrial accidents based on the construction costs. The chi-squared test results indicated that the *p*-value was less than 0.001, indicating a significant difference in the proportion of industrial accident causes based on the construction costs. Among the twenty-three causes, except for four causes, including 'Molds and

supporting post', 'Work environment, natural phenomena such as atmospheric conditions, etc.', 'Fragments, debris, waste', and 'Means of air, water transportation', it was analyzed that accidents caused by all other causes were more frequent on construction sites with construction costs of less than KRW 5 billion. Next, a chi-squared test was performed to verify the frequency differences of accident types based on construction costs. The analysis revealed that the p-value was less than 0.001, indicating a significant difference in the proportion of industrial accident types based on the construction cost. Among the seventeen accident types, except for three types (struck by object, oxygen deficiency, and abnormal temperature contact), it was analyzed that all the other types had a higher frequency on construction sites with construction costs of less than KRW 5 billion.

### 4.1.2. Analysis of Critical Causes and Accident Types in Small-Scale Construction Sites

Considering the stark disparity in the frequency of industrial accidents, based on their causes and accident types between small-scale and medium-/large-scale construction sites, we undertook a comprehensive analysis. Figure 4 delineates the principal causes of fatal accidents in small-scale construction sites. The absolute frequency (AF) and relative frequency (RF) for scaffolding and working platforms emerged as the highest, with AF = 190 and RF = 0.725. This underscores that scaffolding and working platforms were the predominant factors leading to fatal accidents in small-scale construction sites. Subsequent to this, the major causes in small-scale construction sites were identified in the following sequence: means of land transportation (AF = 138, RF = 0.734), stepped structure and opening (AF = 128, RF = 0.677), and stairs and ladders (AF = 96, RF = 0.813).

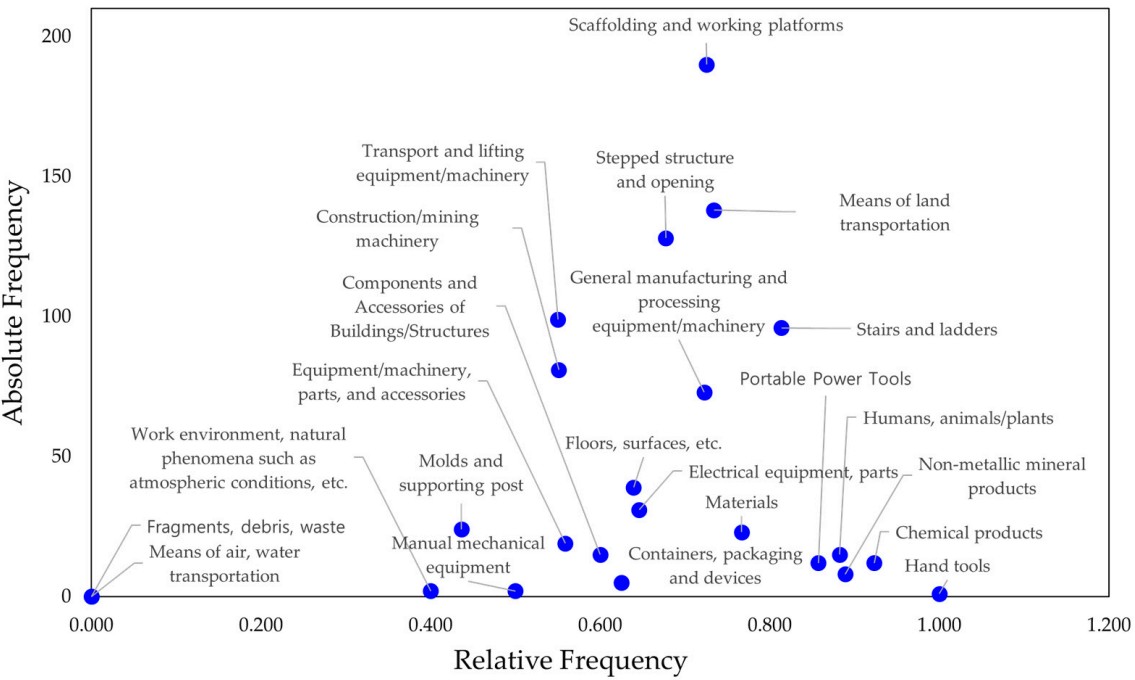

**Figure 4.** Critical causes in small-scale construction sites.

The results of the analysis of critical accident types in fatal accidents at small-scale construction sites are indicated in Figure 5. The analysis revealed that fatal accidents caused by falls had an overwhelmingly higher absolute frequency and relative frequency than other accident types (AF = 547, RF = 0.710). Furthermore, fatal accidents resulting from collisions were also relatively high compared to other fatal accidents (AF = 109, RF = 0.619). Following collisions, accidents involving being pressed under or overturned (AF = 73, RF = 0.709), being struck by an object (AF = 59, RF = 0.496), and being caught in between (AF = 45, RF = 0.600) were identified as critical factors contributing to fatal accidents.

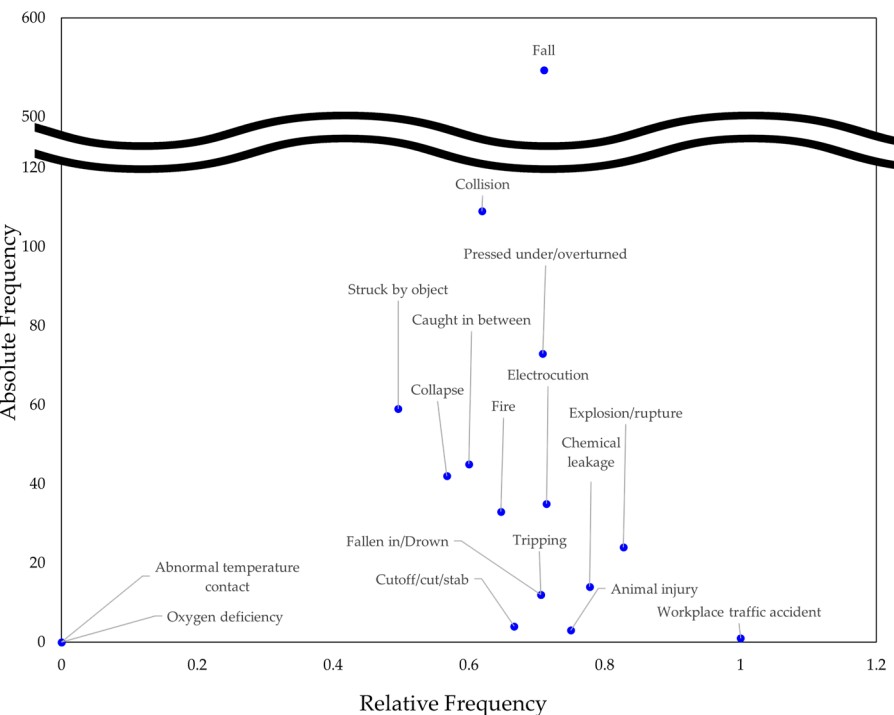

**Figure 5.** Critical accident types in small-scale construction sites.

The analysis of the most critical causes and accident types in fatal accidents at small-scale construction sites revealed that the most critical cause was scaffolding and working platforms, while the most critical accident type was falls.

*4.2. Presentation of Accident Trends in Small-Scale Construction Sites*

An analysis of the accident trends was conducted regarding fatal accidents occurring at small-scale construction sites, specifically those caused by the critical factors of scaffolding and working platforms as well as falls. The scope of the analysis encompassed industrial accident cases reported to the Occupational Safety and Health Agency from 2018 to 2022. In small-scale construction sites, there were 190 work-related fatal accidents due to scaffolding and work platform accidents and 547 work-related fatal accidents due to falls. The trends in fatal accidents were identified through LDA topic modeling and network analysis applied to the overview of the industrial accident data.

It was recognized that upon an initial review, the network analysis figures for each topic might have appeared to have overlapping characteristics. This perception was believed to stem from the broader categorization of accident types. However, upon closer examination, distinct differences in keyword interrelations across these networks were observed. These subtleties, while nuanced, were deemed instrumental in understanding the underlying patterns and causative factors of accidents in unique contexts. The decision to present these topics in detailed individual figures, instead of consolidating them, was based on the commitment to ensure that these critical distinctions were clearly communicated. By delving into the granular details of each topic, a comprehensive overview was intended to be provided, ensuring that the intricate relationships and patterns intrinsic to each accident type were distinctly captured and interpreted. A holistic representation of the findings was considered paramount to convey the multifaceted nature of accidents in small-scale construction sites.

4.2.1. Topic Modeling and Network Analysis for the Most Critical Cause

Accident trends related to work-related fatal accidents caused by scaffolding and working platforms in small-scale construction sites were analyzed using LDA topic modeling and network analysis. Perplexity and coherence scores were calculated on the pre-tokenized

data to determine the optimal number of topics for fatal accidents related to scaffolding and working platforms (Figure 6). Based on the calculated perplexity and coherence score, the optimal number of topics was determined to be five.

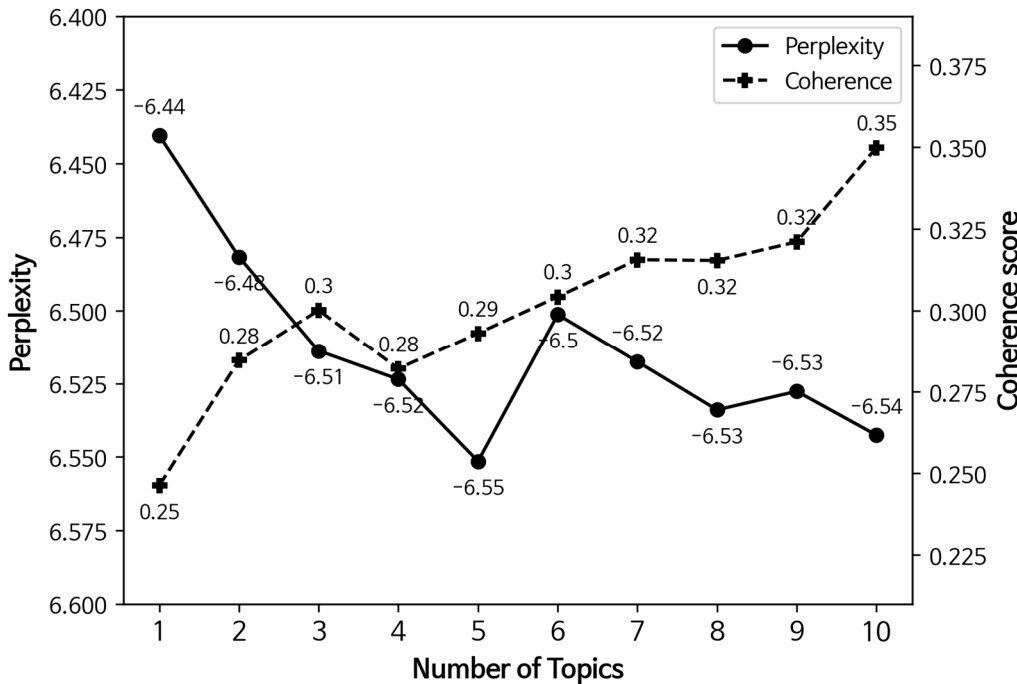

**Figure 6.** Perplexity and coherence score in accidents caused by scaffolding and working platforms.

An inter-topic distance map (IDM) of the topic modeling results is presented in Figure 7. The IDM visualizes the relationships and distances between different topics derived from the LDA topic modeling. In the IDM shown in Figure 7, numbers 1 through 5 inside the circles represent the topic numbers. The size of each circle reflects the prevalence of the corresponding topic: larger circles indicate topics that are more prevalent in the dataset. The spatial distance between circles on the map illustrates the relevance between topics: topics that are closer to each other are more closely related in terms of content, while those farther apart are less related. The IDM represents the entire set of topics from the learned topic model on a two-dimensional scale [26]. Topics are depicted as circles, where larger circles indicate a higher prevalence. Closer distances between topics indicate a higher relevance, while greater distances imply a lower relevance [27]. The analysis of the IDM revealed that there was no overlapping area between topics, and the distribution of data for each topic was appropriately balanced: Topic 1—48 incidents, topic 2—31 incidents, topic 3—29 incidents, topic 4—41 incidents, topic 5—41 incidents.

Through LDA topic modeling, it was analyzed that accidents in small-scale construction sites caused by scaffolding and working platforms could be broadly classified into five distinct types. Next, network analysis was performed to understand the accident trends considering the interactions between factors within each topic. The results of the network analysis for topic 1 are indicated in Figure 8. The analysis of topic 1 revealed that 20 nodes and 50 edges were confirmed. The average degree of connectivity was 5.0, and the network density was about 0.26. The important nodes identified included 'Scaffolding', 'Floor', 'Apartment', 'Rope', 'Outer wall', etc. In particular, the high frequency of 'Scaffolding' and 'Floor' indicates that they were likely to be the main causes of fatal accidents when scaffolding work was performed with unstable platforms or inadequate floor conditions. Furthermore, the connections with 'Apartment', 'Rope', and 'Outer wall' point toward potential risk factors during rope or outer wall operations at apartment construction sites.

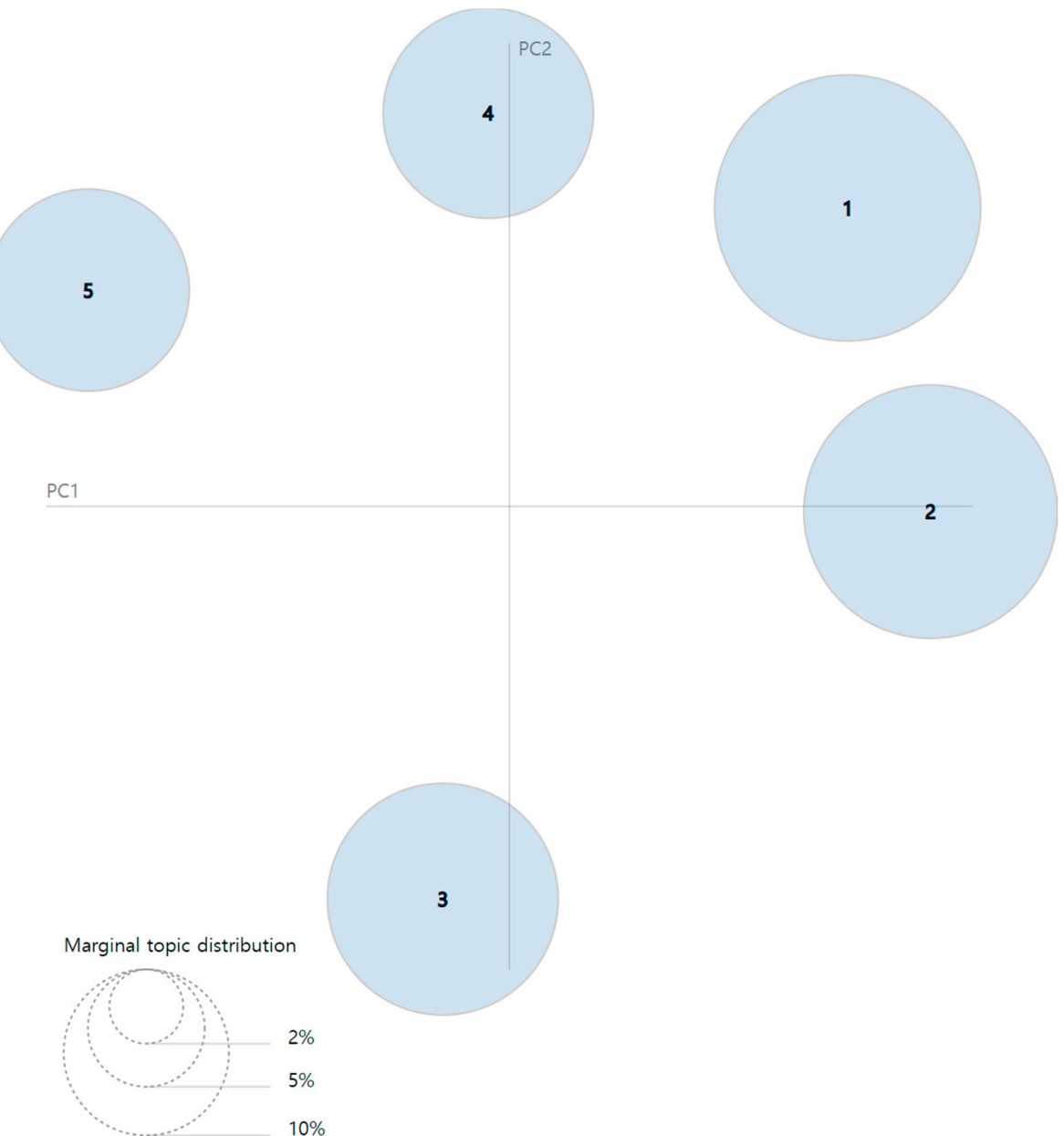

**Figure 7.** Inter-topic distance map of accidents caused by scaffolding and working platforms.

The results for topic 2 regarding accidents caused by scaffolding and working platforms in small-scale construction sites are indicated in Figure 9. In topic 2, the main factors related to fatal accidents involving scaffolding and working platforms in small-scale construction sites were analyzed. For the network presented, 21 nodes and 50 edges were confirmed, exhibiting an average degree of connectivity of 4.76 and a network density of approximately 0.24. The important nodes identified included 'Installation', 'Floor', 'New construction', 'Footing', and 'Scaffolding'. In particular, the frequency between 'Installation' and 'Floor', as well as 'New construction', was notably high. This indicates that the condition of the floor during the initial stages of scaffolding or platform installation in new construction buildings can have a significant impact. The analysis of accident trends revealed that fatal accidents were more likely to occur during the early stages of working platform installation on new construction buildings with inadequate floor conditions. These results emphasize the importance of enhancing safety inspections during the installation of platforms and scaffolding at construction sites.

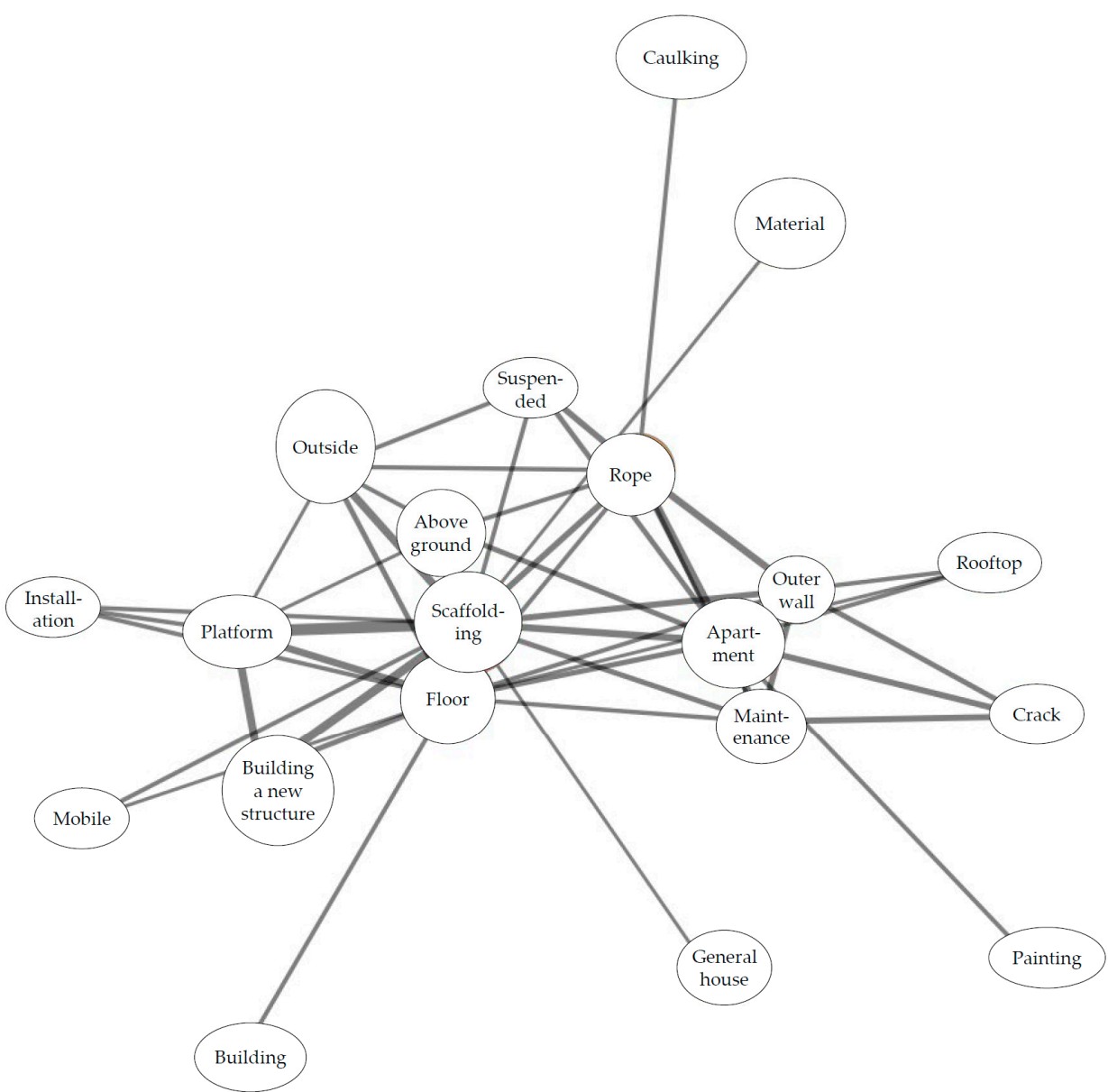

**Figure 8.** Network analysis results for topic 1 in scaffolding and working platforms accidents.

Additionally, it is worth mentioning that Figure 9 also illustrates a secondary network consisting of the nodes 'Apartment' and 'Rooftop', which appear separate from the primary network. This secondary network emerged from our analysis but was deemed less directly related to the central topic of our study. While we did not provide an extensive interpretation for this secondary network in the main content, its inclusion in the figure serves to ensure transparency and provide a complete view of our analytical results.

The network analysis of topic 3 identified complex causes and interactions of fatal accidents related to scaffolding and working platforms in small-scale construction sites (Figure 10). In the analysis, 19 nodes and 50 edges were identified, displaying an average degree of connectivity of about 5.26 and a network density of approximately 0.29. The important nodes identified included 'Mobile', 'Scaffolding', 'New construction', 'Floor', 'Platform', and 'Concrete'. In particular, the high connectivity frequency between 'Scaffolding' and 'New construction' indicates the risks of fatal accidents occurring during scaffolding work. These results were generally similar to topic 2. However, considering the strong connections involving 'Mobile', 'Scaffolding', and 'Platform' in topic 3, it can be inferred that accidents related to mobile scaffolding and working platforms were em-

phasized. It was analyzed that the nodes 'Apartment' and 'Rooftop' were connected only to each other and not to other nodes. This could have two main interpretations: first, the connection between 'Apartment' and 'Rooftop' may signify accidents occurring under specific circumstances or conditions. In this case, these two nodes can be interpreted as related only under specific conditions and insignificant in more general situations. The second possibility is data incompleteness. The lack of connections between 'Apartment', 'Rooftop', and other nodes could imply that there might not have been enough data collected for these nodes. The analysis of accident trends associated with topic 3 revealed that in the case of a newly constructed building with mobile scaffolding installed and the concrete floor being unfinished, the instability of the working platform follows, resulting in an increased risk of fatal accidents. This analysis emphasizes the importance of safety checks for mobile scaffolding and working platforms, particularly re-evaluation after concrete work and floor conditions are addressed.

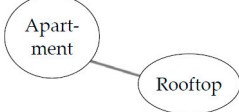

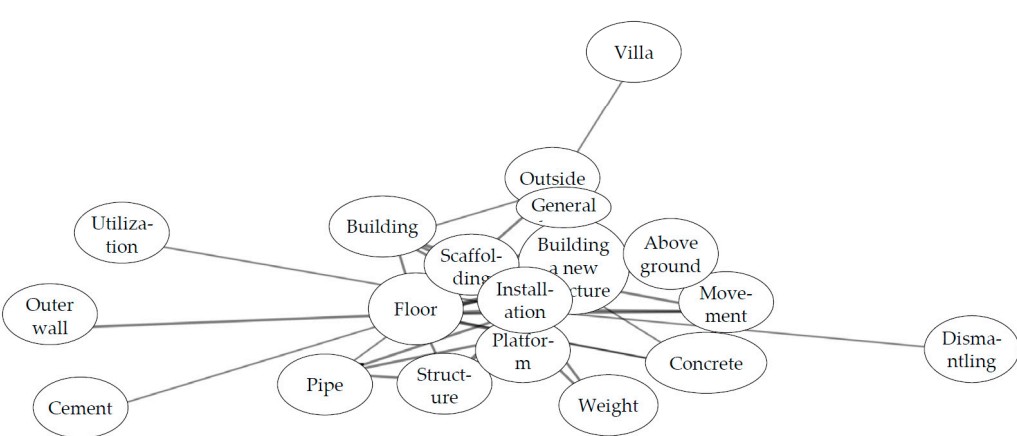

**Figure 9.** Network analysis results for topic 2 in scaffolding and working platforms accidents.

Figure 11 shows the analysis of various causes of fatal accidents related to scaffolding and working platforms in small-scale construction sites corresponding to topic 4. In the network containing 26 nodes and 50 edges, the average degree of connectivity was about 3.85, and the network density was about 0.154. The important nodes identified included 'Scaffolding', 'Floor', 'Platform', 'Outside', and 'Handrail'. Among these, 'Scaffolding', 'Floor', and 'Platform' overlapped with other topics, while 'Outside' and 'Handrail' were found to play a unique role only in topic 4. The analysis of accident trends revealed that external environmental factors (e.g., strong winds) and the absence or instability

of handrails could increase the instability of working platforms, thus elevating the risk of fatal accidents. These results emphasize the importance of safety checks for external environmental factors and proper handrail installation. Furthermore, a node labeled 'Subcontracting' was identified, which was unlike the other topics. This result indicates that fatal accidents related to scaffolding and working platforms are also frequent among subcontracted workers.

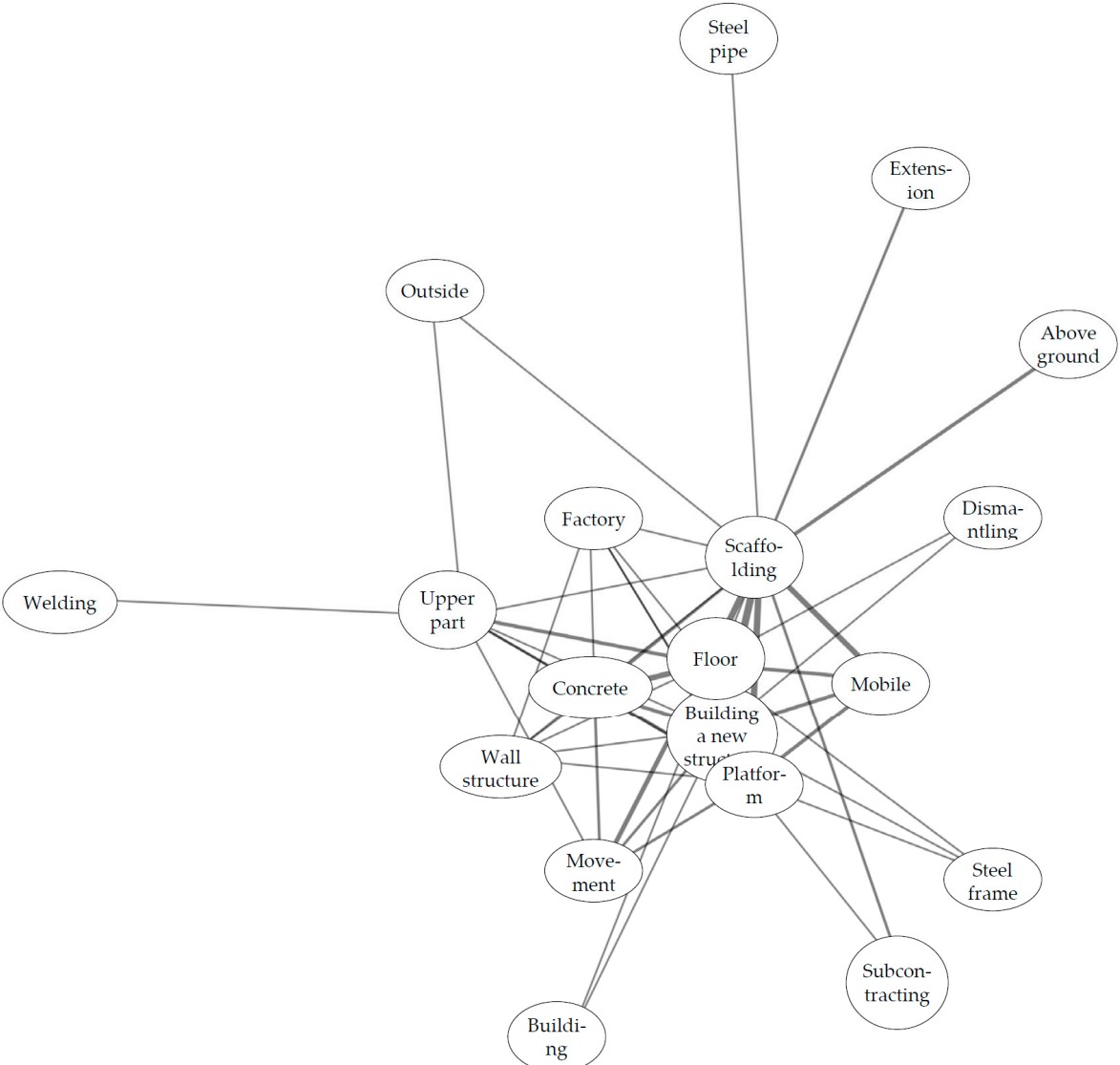

**Figure 10.** Network analysis results for topic 3 in scaffolding and working platforms accidents.

The results of the network analysis for topic 5 are presented in Figure 12. This network consisted of 20 nodes and 50 edges, with an average degree of connectivity of 5.0 and a network density of about 0.26. The important nodes identified included 'Outer wall' which refers to the external facade or surface of a building or structure, 'Floor', 'Scaffolding', 'Rope', and 'Installation'. In particular, the high centrality of 'Outer wall' and 'Rope' elevated the risk of accidents related to work on the outer wall and the use of ropes. The analysis of the accident trends revealed that during work on the outer wall using scaffolding and rope, instability in the rope or installation errors could lead to an increased instability of working platforms, thereby increasing the risk of fatal accidents. These results emphasize the importance of thorough safety checks for ropes when using scaffolding for operations on the outer wall.

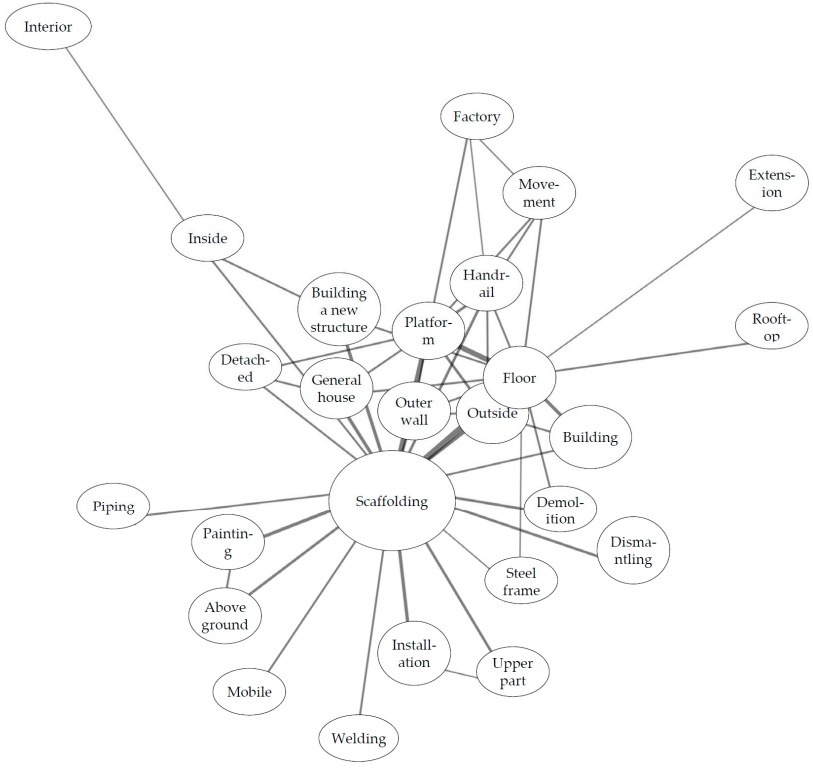

**Figure 11.** Network analysis results for topic 4 in scaffolding and working platforms accidents.

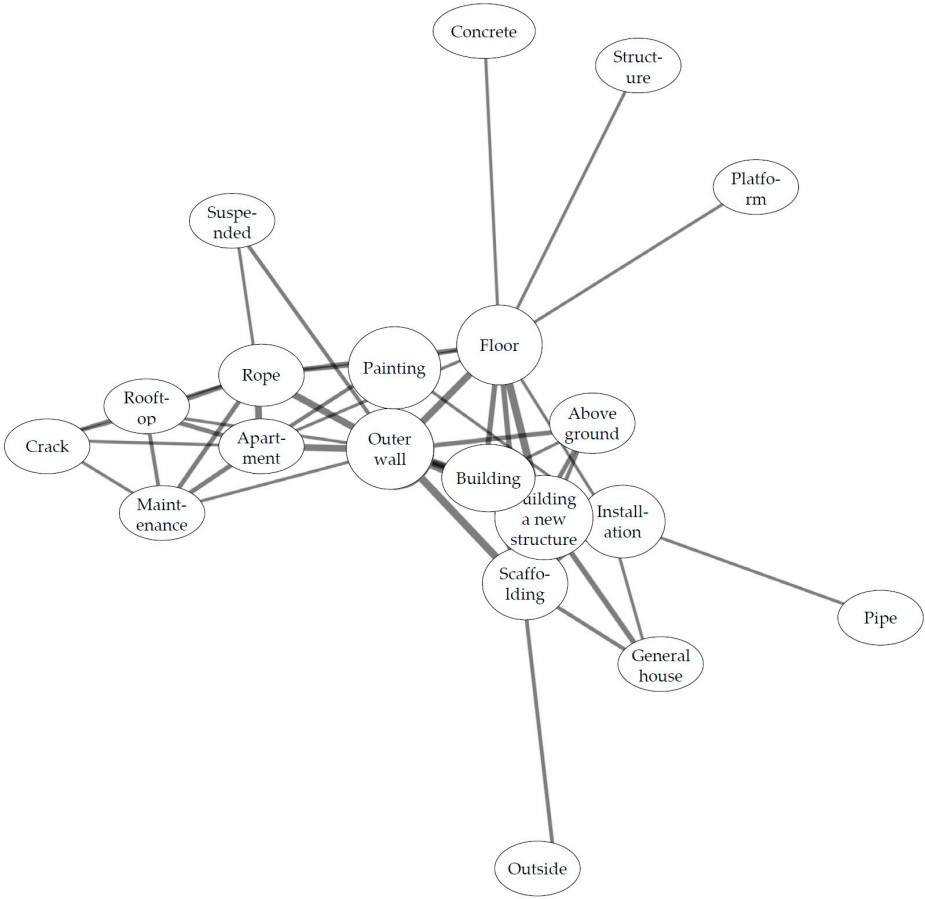

**Figure 12.** Network analysis results for topic 5 in scaffolding and working platforms accidents.

4.2.2. Topic Modeling and Network Analysis for the Most Critical Accident Type

The trends of accidents caused by falls in work-related fatal accidents at small-scale construction sites were analyzed using LDA topic modeling and network analysis. Perplexity and coherence scores were calculated on the pre-tokenized data to determine the optimal number of topics for fatal accidents related to falls (Figure 13). Based on the calculated perplexity and coherence score, the optimal number of topics was seven.

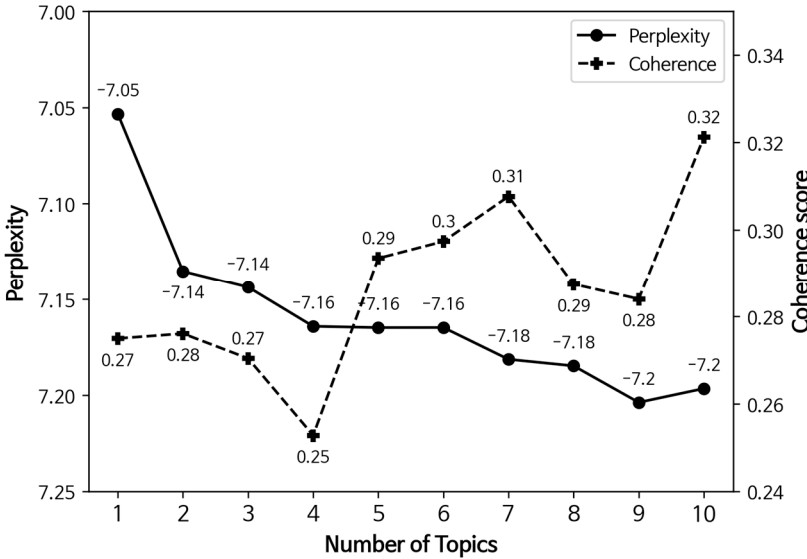

**Figure 13.** Perplexity and coherence score of fall accidents.

The IDM for accidents caused by falls is presented in Figure 14. In the IDM shown in Figure 14, numbers 1 through 7 inside the circles represent the topic numbers. The analysis of the IDM revealed that there was no overlapping area between topics except for topics 2 and 7. While there was some overlapping area between topics 2 and 7, the extent of the overlap was relatively limited. In terms of the amount of data for each topic, topics 6 and 2 had relatively more data compared to the other topics, and topic 3 appeared to have less data: topic 1—86 incidents, topic 2—110 incidents, topic 3—49 incidents, topic 4—63 incidents, topic 5—56 incidents, topic 6—122 incidents, topic 7—61 incidents.

Through LDA topic modeling, it was determined that accidents in small-scale construction sites caused by falls could be broadly classified into seven distinct types. Next, network analysis was performed to understand accident trends considering seven interactions between the factors within each topic.

The results of the network analysis for topic 1 are presented in Figure 15. A network analysis focused on topic 1, which pertained to accidents caused by falls at small-scale construction sites, was conducted. Through this analysis, the interaction between key connecting factors and their importance was identified. The highest frequency of connections was observed between 'Floor' and 'New construction'. In other words, this suggests that accidents involving falls from heights to the ground were frequent at construction sites, particularly in new construction sites. Specifically, accidents falling under topic 1 involved incidents where individuals fell from 'Scaffolding', 'Working platforms', 'Molds', and 'Openings' during the construction of facilities such as 'Neighborhood living facilities', 'Detached houses', and 'Apartments'. Operations frequently associated with fall accidents falling under topic 1 included brick masonry, working while moving, and operations involving the use of ladders. Topic 1 was considered to be the most common type of falling accident, and to prevent accidents related to topic 1, thorough safety checks during floor work in newly constructed buildings are crucial. In particular, ensuring stability before starting work is a crucial measure for preventing "Fall" accidents, which can be concluded

from the analysis. The network of topic 1 consisted of 29 nodes and 50 edges, with an average degree of connectivity of about 3.45 and a network density of about 0.123.

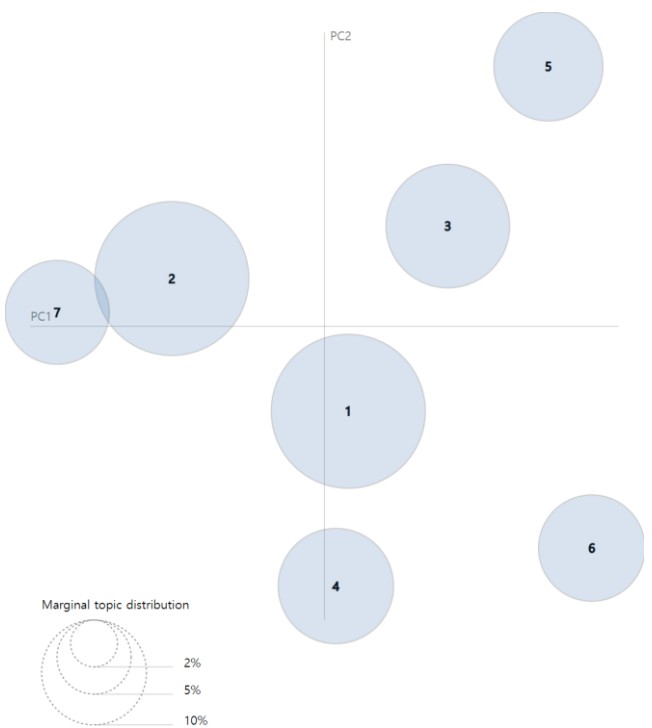

**Figure 14.** Inter-topic distance map of fall accidents.

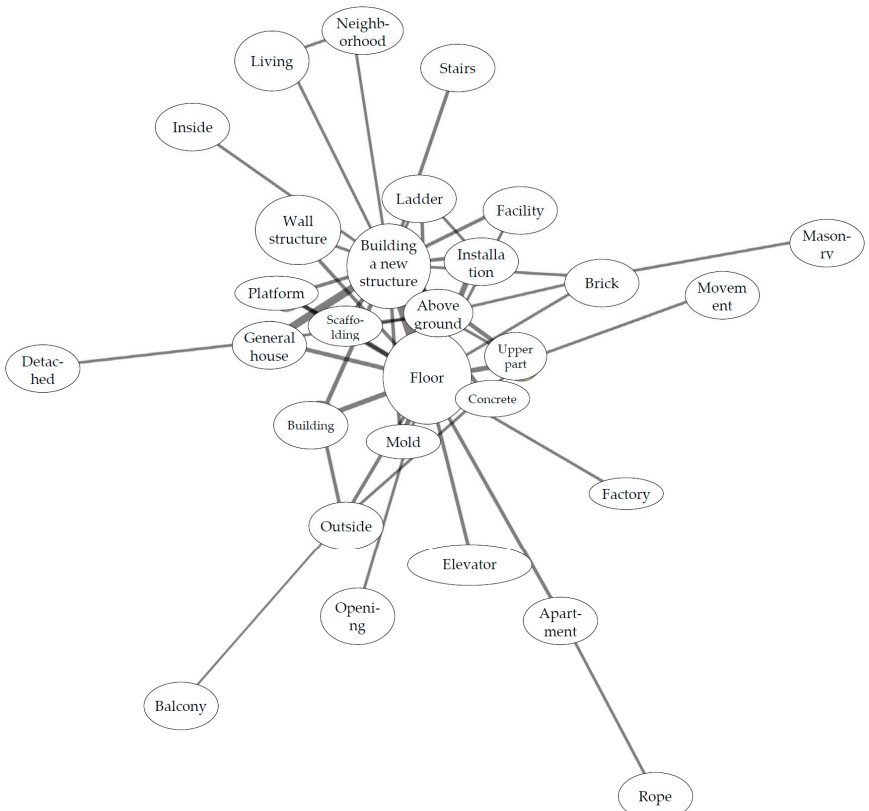

**Figure 15.** Network analysis results for topic 1 in falling accidents.

The network for topic 2 consisted of 20 nodes and 50 edges, with an average degree of connectivity of 5.0 and a network density of about 0.263 (Figure 16). According to the results of the network analysis, fatal accidents involving falls from neighborhood living facility construction sites or general house construction sites, similar to topic 1, formed the central focus. However, in topic 2, a higher frequency of connections was observed between 'Scaffolding', 'Working platforms', 'Outside', and 'Installation'. This indicates that accidents involving falls during the construction process of new buildings where scaffolding and working platforms were used or during the installation process of this equipment were frequent. These accident trends highlight the risks associated with the installation and usage of scaffolding, indicating an increased risk of falling accidents if the installation status and the safety of workers using scaffolding and working platforms are not adequately considered. Therefore, to prevent falling accidents related to topic 2, rigorous safety inspections and management are necessary during the installation and usage of scaffolding and working platforms.

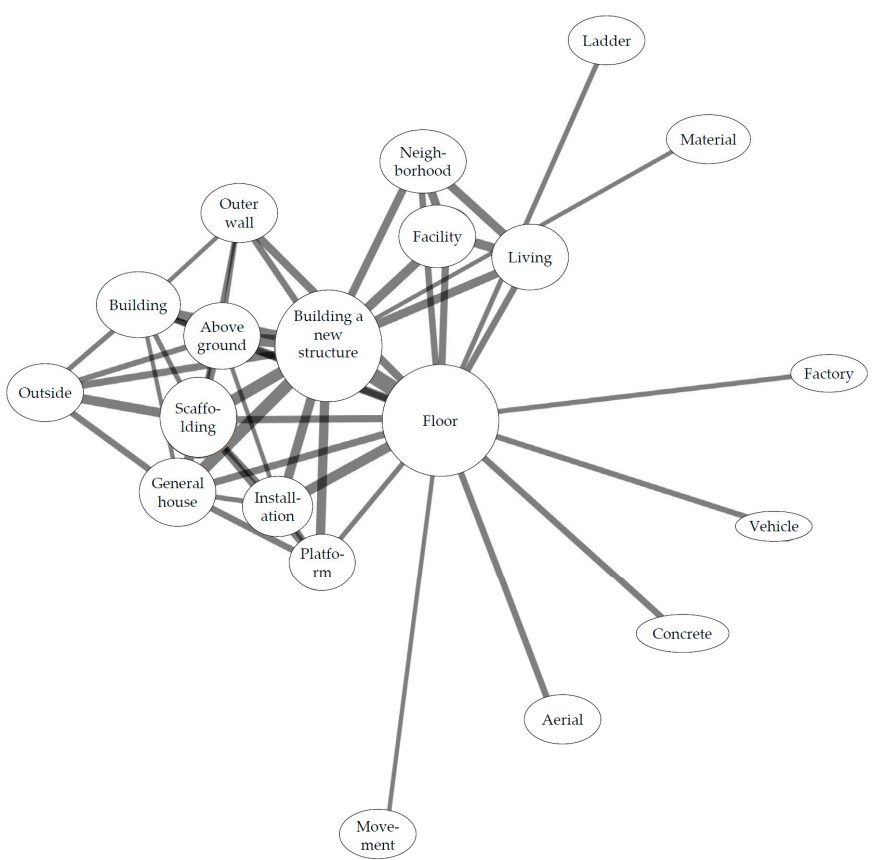

**Figure 16.** Network analysis results for topic 2 in falling accidents.

The results of the network analysis for topic 3 are presented in Figure 17. The network for topic 3 consisted of 25 nodes and 50 edges, with an average degree of connectivity of 4.0 and a network density of about 0.167. For topic 3, it was determined that accidents involving falls during the installation of ladders, scaffolding, working platforms, and equipment were found to be predominant.

The network analysis results for topic 4 are indicated in Figure 18. The network of topic 4 consisted of 22 nodes and 50 edges, with an average degree of connectivity of about 4.55 and a network density of about 0.216. Unlike the other topics, topic 4 indicated that accidents involving falls from ladders during work were a major topic, in addition to accidents occurring at construction sites of new buildings. It was determined that accidents involving falling from ladders frequently occurred during the maintenance and inspection of infrastructure such as roads and bridges. In general, during maintenance or inspection

of bridges, there are often multiple instances where site managers are absent and a small number of workers are involved. Due to these factors, safety and health management may be lacking compared to regular construction sites, creating an environment that is more vulnerable to falling accidents than regular construction sites. However, considering these accidents were identified as the major topics of falling accidents occurring at small-scale construction sites, it is necessary to enhance occupational health and safety management at such sites.

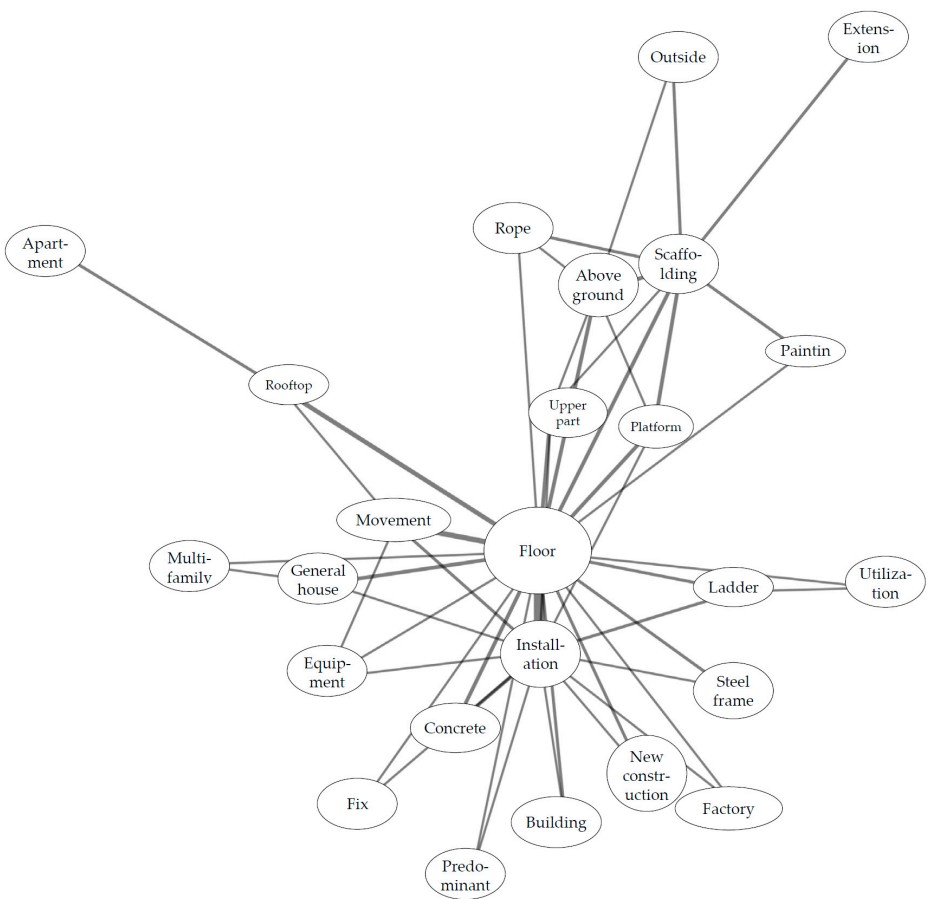

**Figure 17.** Network analysis results for topic 3 in falling accidents.

The results of the network analysis for topic 5 are presented in Figure 19. This network consisted of 22 nodes and 50 edges, with an average degree of connectivity of about 4.55 and a network density of about 0.216. The core keywords of topic 5 were 'Aerial', 'Boarding', 'Installation', 'Vehicle', 'Work platform', 'Loading', 'Painting', and 'Outer wall'. Analyzing the network structure of these keywords revealed that accidents related to vehicles and other equipment were a major type of accident during aerial operations. In particular, the connection structure between 'Boarding' and 'Vehicle', as well as 'Loading', indicates high risks associated with boarding and loading processes on vehicles or other mobile equipment. The connection between 'installation' and 'Outer wall' points to falling accidents during outer wall installation work. Furthermore, the connection structure between 'Painting' and 'Suspended scaffolding' signifies falling accidents while performing painting works using suspended scaffoldings. Through this connectivity structure, the risks associated with operations such as aerial operation, particularly involving the boarding and loading of vehicles and equipment and painting operations, were recognized on construction sites. Through this analysis, it can be concluded that accidents related to vehicles and equipment during aerial operation in construction sites were a major accident type in topic 5. Therefore, the safe usage of vehicles and equipment, providing worker education and

training, and conducting thorough safety checks during operations are necessary to prevent such accidents.

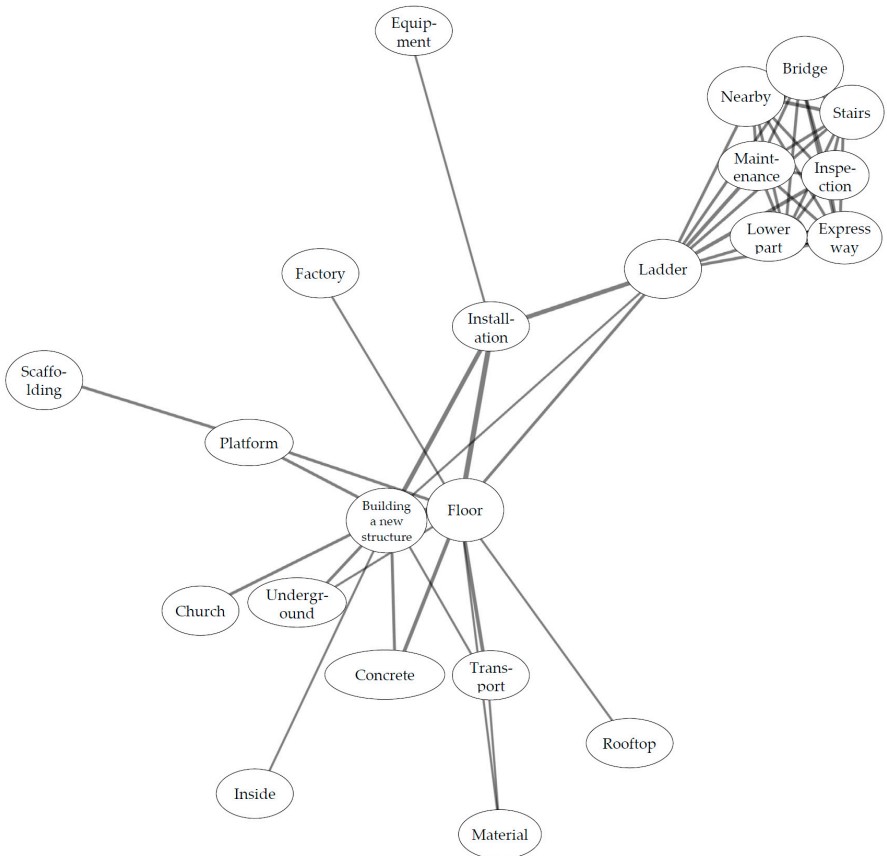

**Figure 18.** Network analysis results for topic 4 in falling accidents.

The network for topic 6 consisted of 23 nodes and 50 edges, with an average degree of connectivity of about 4.35 and a network density of about 0.198 (Figure 20). According to the results of the network analysis, the core accident patterns in topic 6 were related to accidents involving scaffolding and working platforms. In particular, accidents related to falls from scaffolding installed on the outer walls of apartments, accidents occurring during the installation of scaffolding and working platforms, and accidents during the demolition and dismantling processes were identified to fall under topic 6. In comparison with the other topics, a characteristic feature of topic 6 was the strong network formation of falling accidents from scaffolding during work on the outer wall of apartments and during demolition and dismantling operations.

According to the results of the network analysis for topic 7, similar to topic 6, accidents involving falls from scaffolding during the installation process on apartment and building outer walls appeared as a major topic (Figure 21). However, for topic 7, in contrast to the other topics, accidents involving falls while using scaffolding during maintenance work were found to be predominant in this analysis. The network for topic 7 consisted of 24 nodes and 50 edges. The average degree of connectivity in the network was about 4.17, and the network density was about 0.181.

The analysis results of topics 6 and 7 highlight the importance of safety and installation processes for scaffolding and working platforms used in work on the outer wall of apartments and in the general construction of buildings. Moreover, these results emphasize the importance of safety during maintenance, dismantling, and demolition work, emphasizing the necessity for proper safety education and thorough safety inspections for workers involved in such operations.

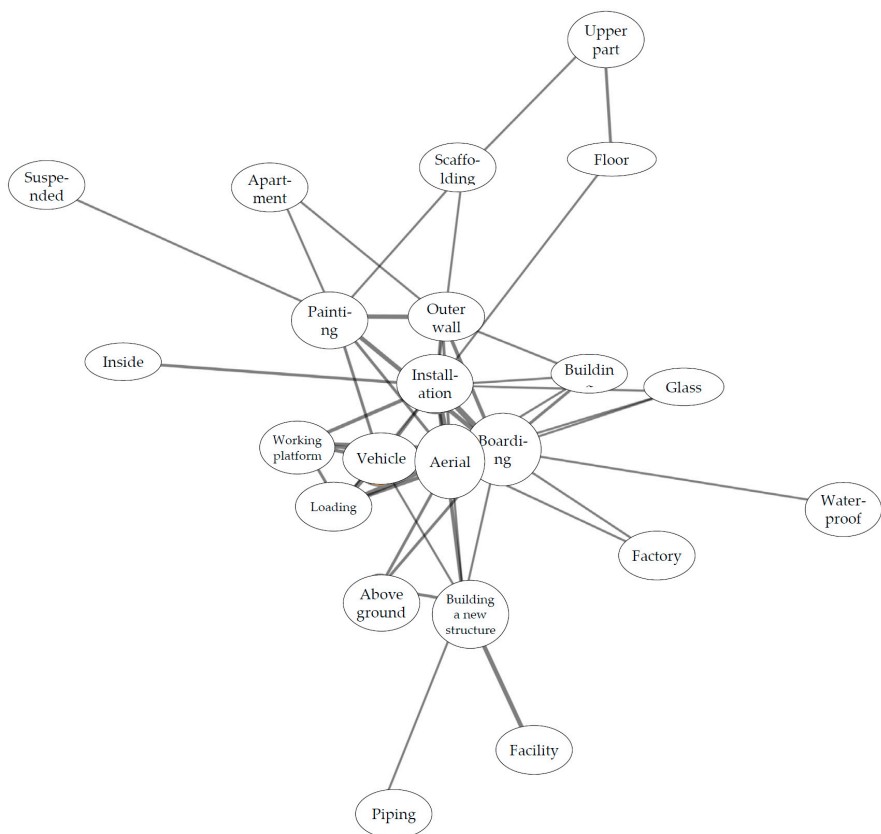

**Figure 19.** Network analysis results for topic 5 in falling accidents.

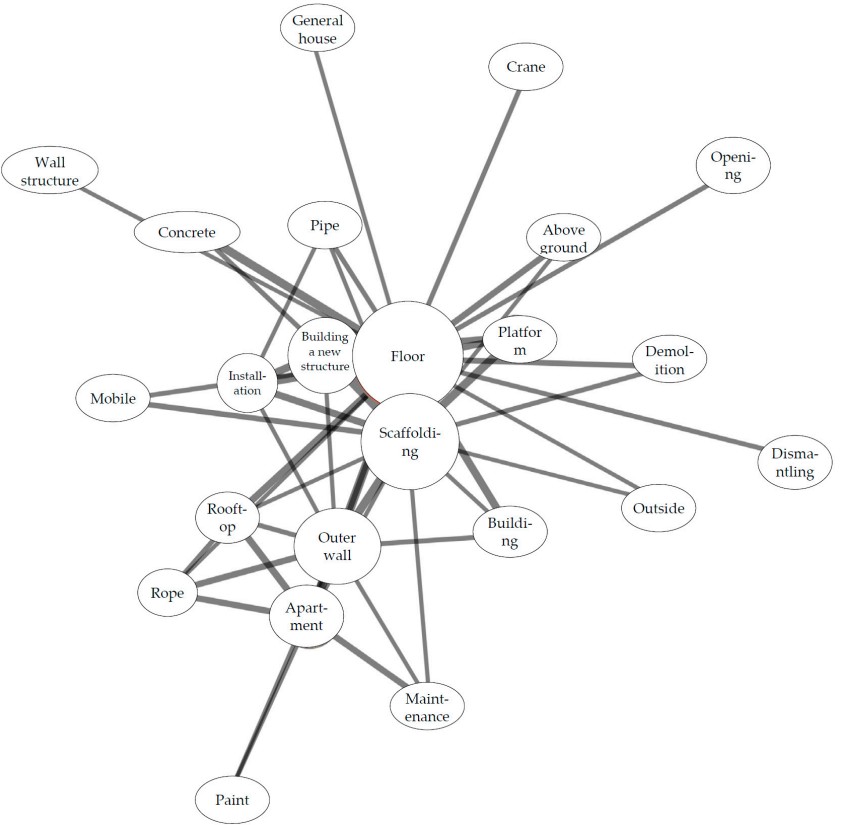

**Figure 20.** Network analysis results for topic 6 in falling accidents.

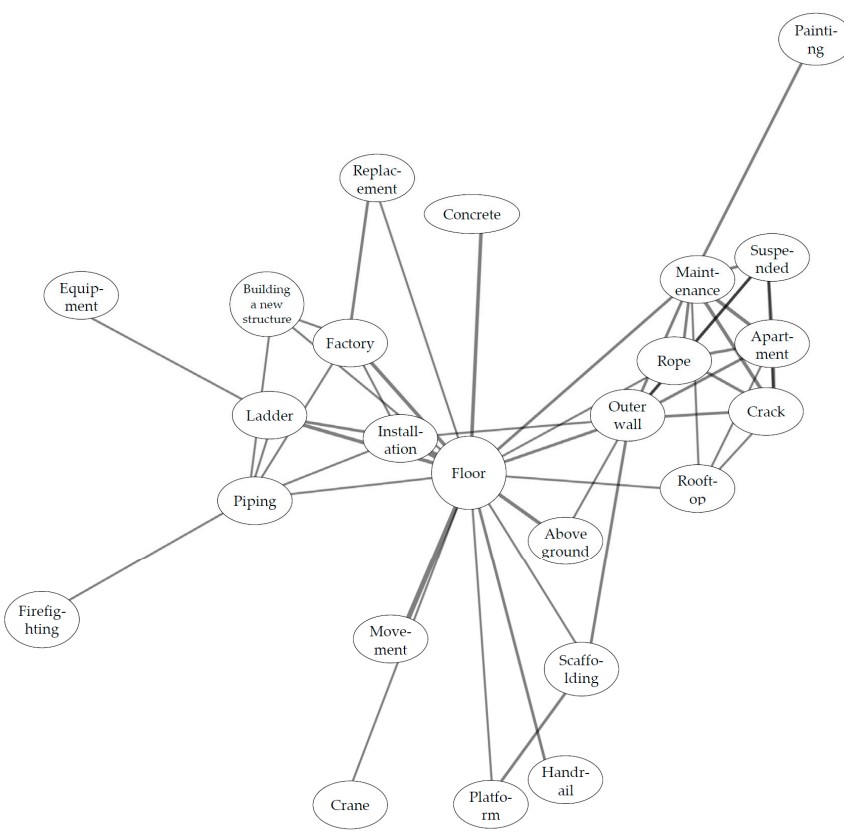

**Figure 21.** Network analysis results for topic 7 in falling accidents.

## 5. Discussion

This study systematically analyzed the causes and trends of industrial accidents in small-scale construction sites in Korea. In particular, the primary objective was to provide detailed accident scenarios for fatal accident causes and how these causes lead to actual accidents. To achieve this objective, various analytical methods were applied in this study. The results aimed to deeply understand the accident causes in small-scale construction sites and provide the scientific basis for establishing effective safety and health management strategies.

The results of this study have revealed several important findings compared to previous studies [5,9]. First, the proportion of fatal accidents in small-scale construction sites was found to be higher than that in medium-/large-scale construction sites. This indicates that workers in small-scale construction sites are exposed to higher accident risks. Second, it was confirmed that the causes and types of fatal accidents differ between small-scale and medium-/large-scale construction sites. This disparity is directly related to the scale of the construction site, allowing for the identification of specific critical accident causes and types that require particular attention in construction sites of a certain scale.

Through this study, the most critical accident cause on small-scale construction sites was identified as 'Scaffolding and working platforms', and the most critical accident type was identified as 'Fall'. For accidents involving scaffolding and working platforms and falling accidents on small-scale construction sites, topic modeling and network analysis were conducted to analyze how the various factors interact to cause fatal accidents and to systematically analyze accident trends. In the analysis of accidents caused by scaffolding and working platforms, various topics were utilized to understand the main causes and patterns of accidents occurring in various situations, such as apartment construction sites, new buildings, and situations involving mobile scaffolding and working platforms. In particular, the analysis focused on aspects such as the connectivity between apartments and rooftops, the risks during the initial stages of installing working platforms in newly constructed buildings, and the risks associated with mobile scaffolding and working platforms.

These results are expected to significantly assist construction site safety managers and workers in recognizing risk factors in specific situations and taking appropriate preventive measures. Furthermore, in the analysis of accidents caused by falling, a detailed analysis was conducted on the main causes and patterns of falling accidents in various situations, including falls in new construction sites, falls during the installation and use of scaffolding and working platforms, and falls during demolition and dismantling work.

Recognizing the importance of delineating the theoretical contribution, managerial implications, limitations, and future research directions in academic research, we are grateful for drawing our attention to this aspect. Our study offers a novel perspective on the safety measures and accident trends in small-scale construction sites, particularly in the Korean context. It bridges the gap in the existing literature by providing a comprehensive analysis using statistical tools, LDA topic modeling, and network analysis. This methodological approach in itself is a significant contribution, as it offers a multidimensional understanding of the data. Our findings can guide construction site managers, policy makers, and safety trainers in designing more effective safety protocols. Furthermore, while our study provides valuable insights into small-scale construction sites in Korea, the generalizability of our findings to larger construction sites or to different cultural contexts might be limited. Additionally, our analysis was based on data from 2018 to 2022, which, although recent, might not capture the very latest trends in the industry. For future research directions, subsequent research can expand on our findings by incorporating qualitative insights, perhaps through interviews with construction site workers and managers. This can provide a richer understanding of the on-the-ground realities and the nuances behind the data. A comparative study with medium- or large-scale construction sites or sites from different countries can further broaden our understanding of construction site safety.

In conclusion, this study systematically analyzed the main causes and patterns of various accidents occurring on small-scale construction sites. While previous studies have generally covered the causes of industrial accidents on construction sites from a general perspective, this study provided a more specific analysis of the causes of industrial accidents on small-scale construction sites. Through this, it offers a deeper understanding of the causes of industrial accidents in small-scale construction sites and their corresponding solutions. These analytical results are expected to greatly contribute to enhancing safety management and preventive measures on construction sites. Furthermore, these study results are expected to contribute to establishing a stronger safety culture within construction sites and to play a crucial role in ensuring the safety of construction workers.

Lastly, this study presents future research directions. First, since this study mainly focused on small-scale construction sites in Korea, analyzing the causes and trends of industrial accidents in different countries or on sites of different scales is necessary. Second, while this study analyzed the causes and trends of industrial accidents, additional time-series analysis is required to understand how these causes and trends change over time. Such future research could complement and expand upon the results of this study.

## 6. Conclusions

This study conducted a systematic analysis on 1511 records from 2018 to 2022, identifying prevalent accident types and causes in small-scale construction sites in Korea using LDA topic modeling and network analysis. In particular, the primary causes of falling accidents on construction sites and the associated risk factors were clearly identified, providing important evidence for establishing prevention and management strategies for falling accidents. These analytical results provide important evidence for enhancing safety management and preventive measures in construction sites.

The results of this study have greatly contributed to a comprehensive understanding of the causes and trends of accidents in small-scale construction sites. Furthermore, these study results are expected to contribute to establishing a stronger safety culture within the construction sites. In particular, through the identified major causes and risk factors

from this study, it will be possible to formulate and implement more systematic safety management strategies in construction sites.

While this study has extensively analyzed the critical factors leading to fatal accidents in small-scale construction sites, focusing specifically on 'scaffolding and working platforms' and 'falling' accidents using LDA topic modeling and network analysis, it's essential to note that the scope was limited to specific scenarios. As a result, validation for the applicability of these results in diverse construction environments becomes crucial [14–17].

In future studies, based on the results identified in this study, it is necessary to analyze the causes and trends of accidents in construction sites of various scales and types. Furthermore, based on the major causes and risk factors identified in this study, the focus should be on developing effective safety management strategies and preventive measures.

It is expected that such follow-up studies guided in this direction will significantly contribute to enhancing overall safety in the construction industry.

**Author Contributions:** Conceptualization, J.-M.H., J.-H.W. and S.-H.S.; methodology, J.-M.H. and S.-H.S.; software, J.-M.H. and H.-J.J.; validation, J.-H.W. and S.-H.S.; formal analysis, J.-M.H. and S.-H.S.; investigation, J.-M.H. and S.-H.S.; resources, J.-M.H.; data curation, J.-H.W. and H.-J.J.; writing—original draft preparation, J.-M.H.; writing—review and editing, J.-H.W. and S.-H.S.; visualization, H.-J.J. and S.-H.S.; supervision, J.-H.W. and S.-H.S.; project administration, J.-M.H., J.-H.W., and S.-H.S.; funding acquisition, J.-M.H. and S.-H.S. All authors have read and agreed to the published version of the manuscript.

**Funding:** This work was supported by the Occupational Safety and Health Research Institute of Korea (2021-OSHRI-520), Chungbuk National University BK21 program (2021) and the Human Resources Development of the Korea Insitute of Energy Technology Evaluation and Planning (KETEP) grant funded by the Korea government (No. 20224000000070).

**Data Availability Statement:** Not applicable.

**Acknowledgments:** Not applicable.

**Conflicts of Interest:** The authors declare no conflict of interest.

## Appendix A

**Table A1.** Classification criteria and definitions for accident cause.

| Main Category Code | Subcategory Code | Cause | Classification Criteria and Definitions |
|---|---|---|---|
| 0 | | Equipment/Machinery | This category includes equipment powered by energy and machinery mechanisms and devices used in specific processes. It includes parts or accessories attached to machinery, forming part of the machinery structure. This classification also includes cases where such equipment and machinery were operating for their intended work purposes and caused an accident during operation. Appliances like computers, audio and video devices, and heating/washing/cooling machines are included. Special-purpose vehicles designed for specific operations rather than transportation purposes are classified under the same code. However, vehicles designed for transporting people, goods, etc., are classified as means of transportation (6). Portable power machinery is classified as portable machinery-power (11). If components were attached to the machine at the time of the accident or it is suspected to have been so, the entire machine is classified as the cause. However, if parts or accessories have been separated from the machine or the entire machine is irrelevant to the accident, parts/accessories (2) are classified as the cause. |

**Table A1.** *Cont.*

| Main Category Code | Subcategory Code | Cause | Classification Criteria and Definitions |
|---|---|---|---|
| | 00 | Equipment/Machinery with Insufficient Information | |
| | 01 | General Manufacturing and Processing Equipment/Machinery | This category classifies machines that process and handle materials such as metal, wood, rubber, plastic, non-metallic minerals, etc., through cutoff, shaping, crushing, etc., to create secondary products. Equipment/machinery in this category refers to industrial equipment/machinery used in various industries, including manufacturing and other sectors. Equipment with general-purpose applications across multiple industries falls under this category. However, specialized equipment used for specific purposes in certain industries is classified separately as specialized process equipment/machinery (02), agriculture, forestry, and fishery equipment/machinery (04), and construction and mining equipment/machinery (05) categories. Inclusions: Bending, Rolling, Shaping Machines; Boring, Drilling, Planing, Milling Machines; Extrusion, Injection, Molding, Casting Machines; Grinding, Polishing Machines; Lathes; Presses (excluding printing); Sawing Machines; Screw Thread and Female Screw Cutting Machines; Laser Cutting Machines, Fluid Pressure Cutting Machines, Spot Welding Machines, etc. Exclusions: Agricultural/Horticultural Equipment (041); Logging/Woodworking Machines (042); Conveying/Handling Equipment (03); Construction/Mining Equipment (05); Food Cutting Machines (02101); Meat Grinders (02102); Paper Machines (022); Textile, Clothing, Leather Production Machines (024); Non-Power Hand Tools (13); Portable Power Saws (11102); Portable Power Surface Finishing Tools (11104) |
| | 02 | Specialized Process Equipment/Machinery | This category classifies equipment and machinery not classified under general manufacturing and processing equipment/machinery. It includes equipment/machinery exclusively used for the production of specific products. |
| | 03 | Transport and Lifting Equipment/Machinery | Machines used for the transportation and handling of specific materials. If it is known whether the related parts were attached to the entire machine during the accident, the entire machine is classified as the cause. However, if the parts are separated from the machine or if the machine is not related to the accident, only the specific parts are classified as the cause. Conveying and handling equipment/machinery are often composed of numerous small parts. For example, hoists, cranes, lifts, and elevators operate using pulleys and wheels. Such parts are classified under codes 22 (Machine Components) and 21 (Electrical Parts). Inclusions: Power Conveyors; Cranes; Hoists; Lifts; Elevators; Jacks Exclusions: Agricultural and Horticultural Equipment (041); Construction and Mining Equipment (05); Logging and Woodworking Equipment (042); Woodworking Machinery (014); Crane Accessories (22399); Electrical Parts (21); Means of Transport (6) |
| | 04 | Agriculture, Forestry, and Fishery Equipment/Machinery | |

**Table A1.** *Cont.*

| Main Category Code | Subcategory Code | Cause | Classification Criteria and Definitions |
|---|---|---|---|
| | 05 | Construction/Mining Machinery | |
| | 09 | Other Equipment/Machinery | |
| 1 | | Portable and Manual Mechanical Equipment | This category includes portable hand tools (both non-powered and powered) and manually operated mechanical equipment. If parts causing the accident were attached to a tool, the entire tool is classified as the cause. However, if the parts were separated from the tool or if the primary purpose and function of the tool were unrelated to the accident, the parts are classified separately. For portable hand tools manufactured with both powered and non-powered capabilities, if it is unclear whether they were powered or non-powered at the time of the accident, they are classified as portable tools with no clear power status (12). Inclusions: Hand Tools; Portable Power Tools; Portable Tools-Unclear Power Status; Ladders; Medical Instruments; Unspecified Sewing Equipment Exclusions: Containers, Utensils, Furniture, and Equipment (4); Tool Storage Boxes (43103); Equipment/Machinery (0); Machine Jacks (03902); Parts, Accessories and Materials (2); Crane Accessories (223); Electrical Parts (21); Waterproof Sheets (24902), Drill Bits (22104), Saw Blades (22106) |
| | 10 | Portable and Manual Mechanical Equipment with Insufficient Information | |
| | 11 | Portable Power Tools | Handheld tools that require an energy source (electricity, gasoline, diesel, coal, air, steam, etc.) for operation and are held in the hand while being used. Portable tools are classified based on their general functions. Inclusions: Nail Guns; Portable Spray Equipment; Stapling Tools Exclusions: Equipment/Machinery (0); Agricultural and Horticultural Equipment (041); Power Lawnmowers (04111); Hair and Hand Dryers (09401); Vacuum Cleaners (09401); Hydraulic/Compressed Air Jacks (03902); Metalworking Machinery (013), Woodworking Machinery (014); Stationary Drills (01203); Stationary Woodworking Circular Saws (01401); Drill Bits (22104), Saw Blades (22106); Hand Tools (13) |
| | 12 | Portable Tools (Unclear Power Status) | This category is used for classifying portable tools designed to be powered and unpowered. It is specifically used when determining whether a portable tool was powered or unpowered at the time of the accident is difficult. Exclusions: Hand Tools; Portable Power Tools |
| | 13 | Hand Tools | This classification encompasses various hand tools operated manually without relying on power sources such as electricity, fuel (gasoline, coal), air, steam, fluids, explosives, etc. Inclusions: Hand Tools for Drilling; Hand Tools for Cutting; Hand Tools for Mining; Hand Tools for Fastening; Hand Tools for Measuring; Hand Tools for Striking; Hand Tools for Surface Treatment; Hand Tools for Cleaning; Crowbars; Hammers; Pitchfork; Rakes; Stapling Tools. Exclusions: Worktables (43107); Crane Accessories (223); Fasteners (Nails, Nuts, Bolts, etc.) (225); Drill Bits (22104); Saw Blades (22106); Portable Power Tools (11); Carts and Wheelbarrows (14102) |

**Table A1.** *Cont.*

| Main Category Code | Subcategory Code | Cause | Classification Criteria and Definitions |
|---|---|---|---|
| | 14 | Manual Mechanical Equipment | |
| | 19 | Other Portable and Manual Mechanical Equipment | |
| 2 | | Parts, Accessories, and Materials | This classification categorizes components of equipment/machinery, parts and accessories of automobiles, subcomponents of buildings/structures, and materials such as metals, non-metallic minerals, wood, plastics, etc.<br>When parts and accessories were involved in accidents independently from the entire machinery/equipment, means of transportation, buildings/structures, and other objects, they are classified as causes.<br>If parts and accessories were attached to machinery, equipment, or means of transportation, the machinery/equipment or means of transportation are classified as causes. Even when parts, accessories, or equipment/machinery were attached, if the primary function of the equipment/machinery was not related to the accident, specific components are classified as causes.<br>Inclusions: Rope; Crane Accessories; Electrical Parts; Metallic and Non-Metallic Mineral Materials<br>Exclusions: Chemical Substances and Chemical Products (5); Containers (4); Furniture (4); Machinery (0); Components of Buildings/Structures (33); Means of Transportation (6) |
| | 20 | Parts, Accessories, and Materials with Insufficient Information | |
| | 21 | Electrical Equipment, Parts | Inclusions: Electrical Parts/Accessories for Machinery, Equipment, and Tools<br>Exclusions: Machinery/Equipment (0); Hand tools (13); Means of Transportation (6) |
| | 22 | Equipment/Machinery, Parts, and Accessories | This classification includes non-attached parts necessary for operating and connecting equipment/machinery. It categorizes products used directly in assembly and other processes without any alteration in form.<br>If it is known whether the related parts were attached to the entire machine during the accident, the entire machine is classified as the cause. However, if the parts were separated from the machine or if the machine was not related to the accident, only the specific parts are classified as the cause.<br>Power transmission devices used in industrial machinery are classified under 221 (Parts and Accessories for Machinery/Equipment), while transmission devices for vehicles are classified under 224 (Parts and Accessories for Means of Transportation).<br>Electrical equipment used in internal combustion engines or automobiles (excluding batteries) is classified under 211 (Electrical Equipment and Parts).<br>Inclusions: Dies, Molds; Chains, Leather, Fabric, V-belt Power Transmission Devices; Drums, Pulleys, Sheaves, Cables, Winches; Engines, Turbines; Clutches; Gears; Rollers. |
| | 23 | Non-metallic Mineral Products | This classification categorizes non-metallic mineral products used directly in the detailed elements of buildings and structures (such as bricks, tiles, etc.) without any alteration in form. It excludes components and accessories of buildings (e.g., bathtubs, toilets). |

**Table A1.** *Cont.*

| Main Category Code | Subcategory Code | Cause | Classification Criteria and Definitions |
|---|---|---|---|
| | 24 | Materials | This classification categorizes products or materials that are not limited to specific equipment/machinery or buildings/structures but can be cut or altered in various forms for general and versatile use. It is used for cases not classified in other codes, where the product exists in its initial state, as raw materials, or as unfinished products. It also includes cases where the product exists as components that have been installed and then dismantled. |
| | 25 | Fragments, Debris, and Waste | |
| | 29 | Other Parts, Accessories, and Materials | |
| 3 | | Buildings/Structures and Surfaces | This classification categorizes components of completed or under-construction buildings, structures (bridges, tunnels, towers, dams, etc.), provisional structures installed for construction purposes, and other components of structures, as well as surfaces like the ground and bedrock. If an entire structure was affected by a disaster in an independent state for fabrication or maintenance (where the material became a cause), it is classified under code 2: Parts, Accessories, and Materials. |
| | 30 | Buildings/Structures and Surfaces with Insufficient Information | |
| | 31 | Scaffolding and Working Platforms | |
| | 32 | Molds and Supporting Post | |
| | 33 | Stepped Structure and Opening | |
| | 34 | Stairs and Ladders | |
| | 35 | Floors, Surfaces, etc. | |
| | 36 | Other Provisional Structures | |
| | 37 | Other Buildings/Structures | |
| | 38 | Components and Accessories of Buildings/Structures | |
| | 39 | Other Buildings/Structures and Surfaces | |
| 4 | | Containers, Utensils, Furniture, and Equipment | This category classifies non-industrial items and equipment, such as containers, packaging, and furniture used for transporting/handling goods. It includes protective gear, leisure/sports and corrective equipment, and wheelchairs. |
| | 40 | Containers, Utensils, Furniture, and Equipment with Insufficient Information | |

Table A1. *Cont.*

| Main Category Code | Subcategory Code | Cause | Classification Criteria and Definitions |
|---|---|---|---|
| | 41 | Containers, Packaging, and Devices | |
| | 42 | Household Utensils and Equipment | |
| | 43 | Furniture and Office Equipment | |
| | 44 | Clothing/Protective Gear and Accessories | |
| | 49 | Other Containers, Utensils, Furniture, and Equipment | |
| 5 | | Chemical Substances and Chemical Products | This category classifies various chemical substances and chemical products in different states, such as liquids, gases, fumes, vapors, and solids. Generally, when the specific name of a chemical substance or its state is known, it is classified under the Chemical Substance code (Subcategories 51–56). When only the state of the product is known, it is classified under the code corresponding to that product (Subcategory 57). Emissions from vehicles, furnace gases, and gases generated from kilns are classified under carbon monoxide (55401), while combustion gases and smoke resulting from fires are classified under flame/fire smoke (81505). Inclusions: Acids; Alkalis; Aromatic and Aliphatic Hydrocarbons; Halogens and their Compounds; Metal Dust and Fumes; Pesticides and Insecticides; Coal, Natural Gas, Petroleum Fuels and their Products; Other Chemical Substances and Chemical Products Exclusions: Metallic Materials (241); Non-Metallic Mineral Materials excluding Fuels (242); Fragments, Splinters, Debris (25) |
| | 50 | Chemical Substances and Chemical Products with Insufficient Information | |
| | 51 | Acids | This category classifies various forms of acids. Inclusions: Acid Gases—Halogens; Inorganic Acids—Halogens; Inorganic Acids—Others; Organic Acids Exclusions: Benzoic Acid and Phenylacetic Acid (Herbicides, etc.) (57104); LSD (Lysergic Acid Diethylamide; Hallucinogen) (57204) |
| | 52 | Alkalis | This category classifies chemical substances known as alkalis, bases, and corrosive agents. For corrosive substances with insufficient information, they are classified under 520. For cement mixtures, mortars, and lime (excluding chlorinated lime), these are classified under 521 with calcium hydroxide and calcium oxide. For ash solutions and their products (wastewater and oven cleaners including ash solution), they are classified under 524 with sodium hydroxide, potassium hydroxide, and potassium carbonate. Chlorinated lime is classified under 542 as chlorine and chlorine compounds. Inclusions: Calcium Hydroxide, Calcium Oxide, Calcium Carbonate, Sodium Carbonate, Cement, Lime, Lithium Hydroxide, Sodium Hydroxide, Potassium Hydroxide, Sodium Carbonate Exclusions: Chlorinated Lime (542); Non-alkaline Oven Cleaners |

| Main Category Code | Subcategory Code | Cause | Classification Criteria and Definitions |
|---|---|---|---|
| | 53 | Aromatic and Aliphatic Hydrocarbons | This category classifies non-halogenated substances such as alcohols, aldehydes, amines, aromatic compounds, ethers, ketones, and peroxides, with chlorine, fluorine, bromine, iodine, and astatine. Halogenated compounds are classified under Subcategory 54: Halogens and Halogen Compounds. Inclusions: Alcohols, Antifreezes, Aldehydes, Aliphatic Amines, Aromatic Compounds, Ethers, Ketones, Peroxides. |
| | 54 | Halogens and Halogen Compounds | This category classifies halogens such as bromine, chlorine, fluorine, iodine, astatine, and their compounds. Compounds containing both fluorine and chlorine are classified under '541 Fluorine and Fluorine Compounds'. Vinyl chloride and polyvinyl chloride are classified under the '57209 Plastics and Resins' code, but molded or extruded plastic products are classified under '245'. Chlorinated hydrocarbons used as pesticides are classified under 57105. Halogen-containing acids are classified under the '51 Acids' subcategory. Inclusions: Bromine and Bromine Compounds, Chlorine and Chlorine Compounds, Fluorine and Fluorine Compounds, Iodine and Iodine Compounds, Carbon Tetrachloride. Exclusions: Halogenated Acids (51); Pesticides (57105), Non-chlorine Bleaching Agents (57203), Vinyl Chloride, Polyvinyl Chloride (57209) |
| | 55 | Other Chemical Substances | This category classifies ammonia and its compounds, cryogenic gases, cyanide compounds, oxygen and specific oxides, sewage and mine gases, methane, sulfur and sulfur compounds, and other chemical substances not classified elsewhere. Inclusions: Ammonia and Ammonia Compounds, Carbon Monoxide, Carbon Dioxide, Cryogenic Gases, Cyanide and Cyanide Compounds, Dry Ice, Methane, Mine Gases, Oxygen and Oxygen Compounds, Sewage Gas, Sulfur and Sulfur Compounds, Sulfur Dioxide |
| | 56 | Metal Particles, Trace Elements, Dust, and Fumes | This category classifies metal dust, particles, and mists, excluding dissolved metals. It also classifies fumes generated during heating/melting/welding processes. Accidents caused by metal radiation are classified under radioactive minerals (24301) or ionizing radiation (81301) codes based on the exposure/contact method. Finished metal products are classified under appropriate codes based on their functionality. Inclusions: Arsenic, Arsenic Compounds, Beryllium, Beryllium Compounds, Cadmium, Cadmium Compounds, Lead, Lead Compounds, Mercury, Mercury Compounds, Aluminum, Aluminum Compounds, Antimony, Antimony Compounds, Iron, Iron Compounds, Magnesium, Magnesium Compounds, Manganese, Nickel, Nickel Compounds, Zinc and Zinc Compounds, Fumes Generated During Welding or Joining Exclusions: Finished Metal Products; Dissolved or Solid-State Metals (241); Radioactive Metals (24301); Coal Dust (57301); Grain Dust (71405); Non-metallic Dust (24201), Ionizing Radiation (81301) |
| | 57 | Chemical Products | This category is used for classification when the substance cannot be categorized under other specific items or when the specific chemical substance is unknown. |

**Table A1.** *Cont.*

| Main Category Code | Subcategory Code | Cause | Classification Criteria and Definitions |
|---|---|---|---|
| | 59 | Other Chemical Substances and Chemical Products | |
| 6 | | Means of Transportation | This category classifies means of transportation that move on land, water, or air and are used primarily for transporting people or goods (e.g., cars, passenger trains) or leisure (e.g., canoes, bicycles, jet skis). Vehicles and machinery used directly for agriculture, construction, logging, mining, manufacturing, etc., are classified under 0 (Facilities and Machinery). When accidents involve means of transportation, if a part of the means of transportation causing the accident was attached to the entire means of transportation, the entire means of transportation is the cause. If parts of the means of transportation were detached or separated, or if the means of transportation was not related to the accident, only specific parts become the cause. In other words, unattached vehicle parts and accessories are classified under 224 (Parts and Components of Means of Transportation) and unattached trailers under 22403. Unattached windshields and windows of means of transportation are classified under 22499, and if the cause was the floor surface of the means of transportation, it is classified under 31307. Inclusions: Means of Air, Water, and Land Transportation; Non-Industrial Transport Vehicles Except for Roads; Means of Railway Transportation Exclusions: Machinery (0), Agricultural/Horticultural Equipment (041), Construction/Mining Equipment (05), Logging Machinery (042), Transport and Lifting Machinery (03), Street Sweepers (09999), Vehicle Parts and Accessories (224), Floor Surface of Means of Transportation (31307) |
| | 60 | Means of Transportation with Insufficient Information | |
| | 61 | Means of Land Transportation | |
| | 62 | Means of Air, Water Transportation | |
| | 69 | Other Means of Transportation | This category classifies means of transportation used outside roads or not powered by internal combustion or other internal engines. |
| 7 | | Humans, Animals/Plants | This category classifies living organisms (including infectious and parasitic organisms) and products made from them. HIV related to work is classified under clause 71603 (Virus). Inclusions: Animals and Animal Products, Raw or Processed Food, Infectious and Parasitic Organisms, Humans-Victims, Humans-Humans Other than Victims, Unprocessed Plants, Trees, Vegetables Exclusions: Chemical Substances (5); Wood (244) |
| | 71 | Humans, Animals/Plants | |

<center>Table A1. *Cont.*</center>

| Main Category Code | Subcategory Code | Cause | Classification Criteria and Definitions |
|---|---|---|---|
| 8 | | Work Environment, Natural Phenomena such as Atmospheric Conditions, etc. | This category classifies the work environment and conditions. Natural phenomena such as atmospheric pressure, temperature, and other atmospheric conditions are classified only in limited cases when they were the only identifiable cause, such as climate, atmospheric conditions, and geographic events (floods, earthquakes, avalanches)<br>Inclusions: Atmospheric Pressure; Avalanches, Landslides; Earthquakes; Fire, Smoke; Floods; High/Low-Temperature Environments; Climate and Atmospheric Conditions; Noise |
| | 81 | Work Environment, Natural Phenomena such as Atmospheric Conditions, etc. | |
| 9 | | Other Causes | |

<center>Table A2. Classification criteria and definitions for accident type.</center>

| Classification Code | Type of Occurrence | Classification Criteria and Definitions |
|---|---|---|
| 01 | Fall (person falls from a height) | Incidents where a person falls from an elevated location such as a building, structure, provisional structure, tree, ladder, etc. |
| 02 | Tripping (person slips or trips) | Incidents where a person slips or falls on a flat surface, sloping surface, stairs, etc. |
| 03 | Pressed Under/ Overturned (object falls or overturns) | Incidents where leaning or standing objects are knocked down and pressed underneath, and construction machinery such as forklifts or other equipment overturns or tips over during operation or movement. |
| 04 | Collision (contact with the object) | Incidents where a person's movement or action results in contact or collision with an object (cause). It also includes situations where objects are set in motion (regularly or irregularly) and collide or make contact due to movement while not detaching from their fixed position. |
| 05 | Struck by object (struck by falling or flying object) | Incidents where a person is impacted by an object that becomes dislodged from a fixed position due to forces like gravity, centrifugal force, inertia, etc., or when material is expelled from equipment or other sources, causing harm to the person. |
| 06 | Collapse (building or piled material collapse) | Incidents where soil, piled material, structures, buildings, provisional structures, etc., collapse entirely or when significant parts break, causing the collapse. |
| 07 | Caught In Between (caught or entangled in machinery) | Incidents where a person gets caught or entangled in machinery due to movement between two objects. It includes situations when caught between objects moving linearly, between rotating parts and fixed components, caught in rotating parts such as rollers, or entangled in rotating parts or protrusions. |
| 08 | Cutoff/ Cut/ Stab | Incidents involving direct contact with sharp objects, such as knives or blades, that result in a part of a person's body being severed or cut. It also includes cases where the body comes into contact with rotating blade parts of saws/cutting tools. |

**Table A2.** *Cont.*

| Classification Code | Type of Occurrence | Classification Criteria and Definitions |
|---|---|---|
| 09 | Electrocution | When a person's body comes into direct contact with parts of electrical equipment or is exposed to induced current, resulting in effects such as muscle contraction, difficulty breathing, ventricular fibrillation, etc. It also includes cases where a person comes into contact with special high-voltage sources or is exposed to arcs due to flash contact, short circuit/mixed contact, etc. |
| 10 | Explosion/Rupture | Explosion refers to a rapid process where a substance undergoes chemical or physical changes, accompanied by the sudden release of heat, sound, and pressure. It can occur within buildings, containers, or the atmosphere, including intentional and unintentional events. Rupture, on the other hand, involves the tearing or bursting of pipes, containers, etc., due to physical pressure without a blast pressure. |
| 11 | Fire | Incidents where the unintentional ignition of combustible materials is due to an ignition source. |
| 12 | Imbalance and Excessive Action | Incidents where sudden and rapid bodily movements/actions, without proper handling of objects, result in an accident. This can also involve situations where excessive muscular force is exerted while handling objects, leading to an accident. |
| 13 | Abnormal Temperature Contact | Incidents where a person is exposed to high- or low-temperature environments or objects. |
| 14 | Chemical Leakage/Contact | Incidents where a person is exposed to hazardous or dangerous substances through leakage, contact, or inhalation. |
| 15 | Oxygen Deficiency | Incidents where a person is exposed to an environment with insufficient oxygen, regardless of hazardous substances, leading to inadequate respiratory function. |
| 16 | Fallen In/Drown | Incidents where a person falls into the water and drowns. |
| 31 | Workplace Traffic Accident | Accidents that occur on roads within the workplace. |
| 32 | Off-site Traffic Accident | Accidents that occur on roads outside the workplace. |
| 33 | Maritime or Aviation Traffic Accident | Accidents related to maritime/aviation activities. |
| 41 | Accidents in Sports Events, etc. | Accidents occurring in work-related recreational or sports events/workshops, gatherings, etc. |
| 42 | Intentional Violence | Intentional or unclearly intentional risky behavior (influenced by drugs, mental disorders, etc.) that causes harm to others through physical violence or assault. This category may also include threats, verbal abuse, and sexual violence. |
| 43 | Animal Injury | Incidents where a worker is injured by an animal (dog/cow/horse, etc.), including cases where a worker is bitten or kicked by an animal. |
| 49 | Others | |
| Z | Unclassifiable | |

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
