# Peer review of "Identifying Critical Factors and Trends Leading to Fatal Accidents in Small-Scale Construction Sites in Korea"

_buildings, doi:10.3390/buildings13102472_

Round 1
Reviewer 1 Report
The keyword of "accident causes and types" would better to be divided into two keywords, "accident causes" and "accident types".
The introduction section is too long and hard to read. I recommend the authors to divide the literature review part from the current introduction, and add a new literature review section.
Besides, in the new literature review section, more sub-sections should be added to make the logical sequence more clear. And more relevant paper focusing on worker safety, like (https://doi.org/10.1061/JCEECD.EIENG-1882), could be introduced and cited.
The table name of the table 1 in the current version is strange. Please check.
Each sub-figure should have a name. Besides, please replace the native words by English names.
The current figure 9 is really confusing. Does it really contain two parts? Please check and revised.
Same problems exist in the figure 14.
The current version is really redundent. As lots of the figures in the results section are in the samely kind. I think some of these figures could be merged or changed into sub-figures. And the long contents could be condensed.
Theoretical contribution, managerial implication, limitation and future direction could be added in the discussion section.
The keyword of "accident causes and types" would better to be divided into two keywords, "accident causes" and "accident types".
The introduction section is too long and hard to read. I recommend the authors to divide the literature review part from the current introduction, and add a new literature review section.
Besides, in the new literature review section, more sub-sections should be added to make the logical sequence more clear. And more relevant paper focusing on worker safety, like (https://doi.org/10.1061/JCEECD.EIENG-1882), could be introduced and cited.
The table name of the table 1 in the current version is strange. Please check.
Each sub-figure should have a name. Besides, please replace the native words by English names.
The current figure 9 is really confusing. Does it really contain two parts? Please check and revised.
Same problems exist in the figure 14.
The current version is really redundent. As lots of the figures in the results section are in the samely kind. I think some of these figures could be merged or changed into sub-figures. And the long contents could be condensed.
Theoretical contribution, managerial implication, limitation and future direction could be added in the discussion section.
Author Response
Dear Reviewer,
We would like to express our profound gratitude for the time, effort, and expertise you dedicated to reviewing our manuscript, "Identifying Critical Factors and Accident Trends in Small-Scale Construction Sites in Korea." Your meticulous feedback, constructive criticisms, and invaluable suggestions have been pivotal in refining and enhancing the quality, clarity, and coherence of our work.
The review process is an essential pillar of academic research, ensuring that the published works meet the highest standards of rigor and relevance. Your detailed and thoughtful comments truly reflect your commitment to this process, and we are genuinely appreciative of your dedication.
For a detailed response to each of your comments, we have provided specifics in the attached file. We sincerely hope this will offer clarity on the revisions made and the rationale behind them.
We understand that the review process is time-consuming and requires significant attention to detail. We deeply regret any inconvenience our initial submission might have caused and appreciate your patience and understanding throughout this process.
Once again, thank you for your invaluable contribution to our research and for guiding us in presenting our findings in the best possible light. We are truly grateful for your role in advancing knowledge in our field and apologize for any added burden our manuscript may have imposed.
Warm regards,
Seung-Hyeon Shin

Reviewer 2 Report
In the present article, "Identifying Critical Factors and Accident Trends in Small-Scale 2 Construction Sites in Korea", authors analyze the causes and trends of industrial accidents at small-scale construction sites, providing important evidence to enhance safety management and preventive measures.
The topic is interesting and the suggestion given to the reader I think could more sensibilize people in safety management in the place of work.
Anyway, some changes have to be performed in order to go ongoing with the pubblication.
In the introduction, at line 77, please give to the reader some examples of safety and health regulations of the country, culture, and construction stakeholders, reporting cases in different country based on these factors. Moreover, complete the information with references.
Line 93. After this part, I suggest you to report also the importance of dispose of dataset population to perform correct studies. In particular, we can use dataset arriving from public institution that analyze population data, but also dataset from social media enterpise (please look here https://doi.org/10.3390/rs15092348).
From line 143 to 170. This part is too longer, please search to synthesize. It's correct to give a background of the metodologies used, but th aim in this part is to focusing on the methods used in the present research.
Figure 14. I advice to separate the figures reported, from a graphical point of view and also for the corresponding explanation.
Line 572. Please insert references.
Line 600-615. I think is better to report directly in the conclusion part.
Line 617-623. Here there is a ripetition of what written before.
Line 638. This is a innovative point or it's is already devellopped in part? if the answer is yes, please report references.
Author Response

(The authors gave the same response as above.)

Reviewer 3 Report
The present study, by utilizing various analytical methods, aims to uncover the critical factors leading to fatal accidents and their underlying scenarios in small-scale construction sites in Korea. The research ultimately seeks to inform more effective safety and health management strategies in the construction industry. My comments for corrections are provided as follows:
1) Line 44: It would be beneficial to provide a more explicit rationale for the choice of comparing the construction industry with the manufacturing sector (1 or 2 sentences).
2) Line 142-172: Τhis section does not seem to belong to the "Materials and Methods" section as it provides essential background information and context regarding the scale of construction sites and their associated safety regulations. It would be more appropriate to place this information in the introduction or, if the introduction is becoming too lengthy, consider creating a separate section specifically dedicated to providing the context and regulatory framework for the study. This will make the paper more organized and easier for readers to follow.
3) Lines 173-255: The subsection titled "Research Scheme" should precede "Data Collection", as it outlines the overall methodology and approach before delving into the specifics of data collection. Ideally, the "Research Scheme" section should be organized in a different way, where its components are its subsections, i.e., data collection, statistical analysis, topic modeling, and network analysis.
4) Line 289: Please provide a caption for the Table.
5) Lines 294-296: Please correct this sentence.
6) Lines 296-297: Since the abbreviations have already been introduced there is no need to spell out the whole term every time thereafter..
7) Line 347, 453: The text in Figures 7 and 14 appears to be too small, making it difficult to discern or understand. Please proceed to make the appropriate changes.
8) Line 366, 381: Sentences should not begin with a numeral.
9) Lines 655-658: While reviewing the table, I found it a bit challenging to comprehend due to the absence of clear lines separating the rows. It would be helpful if lines could be added to separate the rows for improved readability.
Author Response

(The authors gave the same response as above.)

Round 2
Reviewer 1 Report
All my suggestions have been tackled, and the quality of the paper could already meet the requirements of the journal. I recommend to accept the paper in the current version.